# Physics-informed neural network for lithium-ion battery degradation stable modeling and prognosis

Fujin Wang [1,2,3], Zhi Zhai [1,2,3], Zhibin Zhao [1,2] ✉, Yi Di[1,2] & Xuefeng Chen[1,2] ✉

Accurate state-of-health (SOH) estimation is critical for reliable and safe operation of lithium-ion batteries. However, reliable and stable battery SOH estimation remains challenging due to diverse battery types and operating conditions. In this paper, we propose a physics-informed neural network (PINN) for accurate and stable estimation of battery SOH. Specifically, we model the attributes that affect the battery degradation from the perspective of empirical degradation and state space equations, and utilize neural networks to capture battery degradation dynamics. A general feature extraction method is designed to extract statistical features from a short period of data before the battery is fully charged, enabling our method applicable to different battery types and charge/discharge protocols. Additionally, we generate a comprehensive dataset consisting of 55 lithium-nickel-cobalt-manganese-oxide (NCM) batteries. Combined with three other datasets from different manufacturers, we use a total of 387 batteries with 310,705 samples to validate our method. The mean absolute percentage error (MAPE) is 0.87%. Our proposed PINN has demonstrated remarkable performance in regular experiments, small sample experiments, and transfer experiments when compared to alternative neural networks. This study highlights the promise of physics-informed machine learning for battery degradation modeling and SOH estimation.

In recent years, the number of lithium-ion batteries is growing at an alarming rate in the whole society, which is an unprecedented impetus to the popularization of renewable energy equipment. With the advantages of high energy density, low self-discharge rate, and long service life[1], lithium-ion batteries have become the main energy storage devices in portable electronic devices, electric vehicles, aerospace, and many other fields[2-8]. In 2019, the global shipments of lithium-ion batteries for new energy vehicles alone reached 116.6 GWh[9]. It is estimated that by 2025, the global lithium-ion batteries installed capacity will reach 800 GWh, and the market value will reach 91.8 billion dollars[10]. The explosive growth of lithium-ion batteries has brought convenience to people's lives, however, its aging and health management have also attracted people's concerns and attention. The aging of lithium-ion batteries is an important issue, and their performance will decline with time until it fails. To ensure long-term, safe, and continuous operation, lithium-ion batteries must be properly maintained and controlled, which includes state-of-health (SOH) assessments. The SOH of a battery is defined as the ratio of the current available capacity to the initial capacity, which can be used as an indicator to measure battery degradation[11]. When the SOH drops to 80%, the battery reaches its first service life. Batteries that have reached their first service life can still be used in fields such as energy

[1]National and Local Joint Engineering Research Center of Equipment Operation Safety and Intelligent Monitoring, Xi'an Jiaotong University, Xi'an 710049 Shaanxi, PR China. [2]School of Mechanical Engineering, Xi'an Jiaotong University, Xi'an 710049 Shaanxi, PR China. [3]These authors contributed equally: Fujin Wang, Zhi Zhai. ✉e-mail: zhaozhibin@xjtu.edu.cn; chenxf@mail.xjtu.edu.cn

storage power stations for secondary utilization. Therefore, it is particularly important to accurately estimate the SOH of the battery.

In recent years, various SOH estimation methods of lithium-ion batteries have been proposed, which greatly advance the development of this field[12–15]. However, accurately estimating SOH is still a challenging problem. Generally, the capacity can be obtained from a complete discharge curve from an upper cut-off voltage to a lower cut-off voltage via the ampere-hour integration, thereby obtaining SOH. In actual application, it is difficult to obtain a complete charge or discharge curve because the battery is rarely fully discharged. Some scholars estimate SOH by establishing battery aging models. Baghdadi et al.[16] proposed a physics-based approach based on Dakin's degradation method to simulate the linear degradation process of batteries. Considering that time-varying temperature conditions have an important impact on the discharge capacity and aging law of lithium-ion batteries, Xu et al.[17] proposed a stochastic degradation rate model based on the Arrhenius temperature model and established an aging model of lithium-ion batteries under time-varying temperature conditions based on the Wiener process. Dong et al.[18] proposed a physics-based model that combines chemical and mechanical degradation mechanisms to predict capacity fade by simulating the formation and growth of solid electrolyte interphase (SEI). Lui et al.[19] proposed a physics-based approach to predict the capacity of lithium-ion batteries by modeling degradation mechanisms such as losses of active materials of the positive and negative electrodes and the loss of lithium inventory.

Given the difficulty in establishing physical models and the difficulty in obtaining complete discharge capacity, many studies have used data-driven methods[20–24] to estimate SOH based on current and voltage curves during charge and discharge. Commonly used data-driven methods include linear regression[25], support vector machines[26], Gaussian process regression[27], deep neural networks[28,29], etc. Xia et al.[30] extracted features from incremental capacity (IC) curves and differential voltage (DV) curves to estimate SOH. Wang et al.[31] extracted valuable health indicators from electrochemical impedance spectroscopy (EIS) as input for Gaussian process regression to estimate SOH. Data-driven methods do not require physical knowledge and only focus on the relationship between input and output, so the extraction of degenerated features is a key part of data-driven methods, which largely determines the performance of the SOH estimation.

However, challenges still stand in the way of developing reliable, accurate, and general SOH estimation methods[14,21]. Physics-based models are stable and accurate, but batteries with different chemical compositions require different model parameters, and the models have high computational costs[32]. The data-driven models have high accuracy and efficiency, but its generalizability depends on the extracted features and have poor stability[14,33]. For instance, due to the high usage variability, existing methods[30,34,35] need to extract specific features for different datasets or different working conditions, leading to the fact that models are dataset-specific, resulting in a waste of computing resources. The promising prospect of physics-informed neural network (PINN)[36,37] lies in amalgamating the strengths of physics-based and data-driven approaches, potentially addressing the aforementioned challenges. Due to the consideration of physical information, PINN can use relatively less data to train the model, and the model is more stable. It is a promising approach in the field of battery prognosis and diagnostics. Aykol et al.[38] classified battery modeling methods that combine physical knowledge and machine learning into five categories, including three Sequential Integration methods, A1–A3, and two Hybrid methods, B1–B2. Among them, an obvious feature of the Sequential Integration method is that the physical model and the machine learning model are standalone, while the Hybrid method fuses the two together. Within this framework, some works has been published[39–43]. Nascimento et al.[39] directly implemented the numerical integration of principle-based governing

equations through recurrent neural networks to simulate the dynamic response of the battery. Wang et al.[42] proposed a battery neural network (BattNN) for discharge voltage prediction based on the equivalent circuit model (ECM). Hofmann et al.[43] used the pseudo-two-dimensional (P2D) Newman model to generate data at different health status points and combined it with experimental data and field data to train the neural network model, which takes advantage of the correlation between internal states and measurable SOH. According to the categories proposed by Aykol et al.[38], these methods belong to the A2[40,41,43] and A3[39,42] categories.

In fact, the Sequential Integration method is relatively straightforward to implement because the physical model and the machine learning model are standalone, making it a practical near-term strategy for battery modeling. Essentially, machine learning models in Sequential Integration method are not subject to physical constraints. The Hybrid methods, on the other hand, are more fundamental as they truly integrate the primary governing equations for battery modeling with data-driven methods. However, due to the complex physical equations contain numerous parameters and are difficult to solve, there are few publications that implement Hybrid methods for SOH estimation. Recent review[38] pointed out that Hybrid methods will become the dominant method in the long term, but it is still an open research question.

In this work, we proposed a PINN for battery SOH estimation, which belong to the B2 architecture. This approach achieves true integration of governing equations and neural networks, resulting in stable and precise SOH estimation. Unlike existing PINN approaches, we also validated its advancements in small sample learning and transfer learning among batteries with different chemistries and charge/discharge profiles. Specifically, first, considering the complexity of the electrochemical equations, it hinders the development of B2-type PINNs. Therefore, we model battery degradation dynamics from the perspective of empirical degradation and state space equations, and utilize neural networks to capture battery degradation dynamics. Second, to make the model more general, we develop a new feature extraction method. The discharge process of a battery is user-specific, and the battery is rarely fully discharged. In contrast, once charging starts, the probability of full charge is high, and it is more fixed and regular. Therefore, we extract features from a short period of data before the battery is fully charged. Third, to verify our method, we carried out battery degradation experiments and released a dataset containing run-to-failure data from 55 batteries. In addition, we also verified our method on other three large-scale datasets with different chemical compositions and charge/discharge protocols, proving the superiority and versatility of our method. We also perform the further task of estimating SOH by transferring degradation knowledge from one dataset to another. These datasets for performing transfer tasks contain batteries with different chemistries and charge/discharge protocols. The results illustrate the effectiveness and generality of the proposed PINN in SOH estimation.

## Results
### Framework overview and flowchart
We developed a PINN for lithium-ion battery SOH estimation, and its flowchart is shown in Fig. 1. Our method is designed for more general, reliable, stable, and high-precision SOH estimation by considering the dynamic behavior of battery degradation as well as the degradation trend.

In the data preprocessing stage (shown in Fig. 1b), statistical features are extracted from a short period of data before the battery is fully charged as the input of the model, which ensures that this period of data exists in most battery datasets, and solves the problem of non-universal features in existing studies. Therefore, our method is applicable to batteries with different chemistries and charge/discharge protocols.

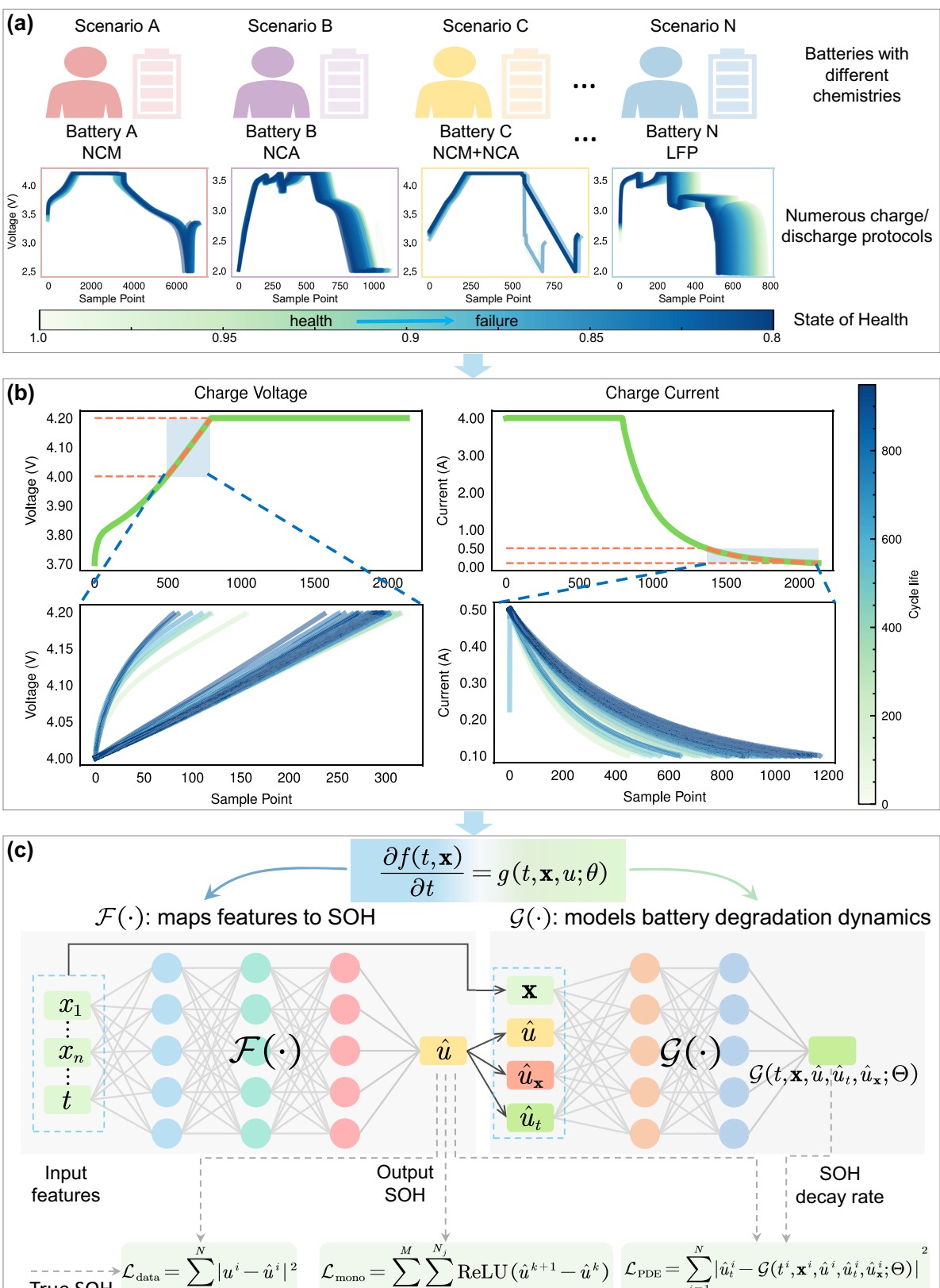

In the SOH estimation stage, due to the complexity of electrochemical equations, there is currently no good way to integrate them with the neural networks. In this work, we modeled the attributes that affect the battery degradation from the perspective of the empirical degradation and state space equation, and utilized neural networks to approximate the established degradation model, effectively achieving the integration of governing equations and neural networks. The proposed PINN consists of two parts: a solution function $f(\cdot)$ that maps features to SOH and a nonlinear function $g(\cdot)$ that models battery degradation dynamic behaviors, as shown in Fig. 1c. The solution $f(\cdot)$, modeling the relationship between features and SOH, is expressed as $u^i = f(t^i, \mathbf{x}^i)$, where $t^i$ represents time, $\mathbf{x}^i$ represents the extracted feature

**Fig. 1 | The flowchart of the proposed PINN for lithium-ion battery SOH estimation. a** The lithium-ion batteries may have different chemistries (e.g., lithium nickel-cobalt-manganate (NCM), lithium nickel-cobalt-aluminate (NCA), and lithium iron phosphate (LFP), etc.). Different users have personalized battery discharge strategies resulting in different degradation trajectories. **b** An illustration of the selected data for feature extraction. We extracted features from a short period of data before the battery is fully charged. These features are used as the inputs of the proposed PINN to estimate SOH. The upper figures are the curves from the 10th cycle, and lower figures are all the curves during the entire life cycle. Aging of the

battery and changes in charge/discharge protocols cause the curves to shift. **c** The structure of the proposed PINN, where $u$ and $\hat{u}$ represent the true and estimated SOH, $t$ and $\mathbf{x}$ represent cycle and features, the superscript $i$ represents sample index, and the subscripts $t$ and $\mathbf{x}$ represent the corresponding partial derivatives. The functions $f(\cdot)$ and $g(\cdot)$ respectively model the mapping between features to SOH and the degradation dynamics of the battery, and $\mathcal{F}(\cdot)$ and $\mathcal{G}(\cdot)$ represent the neural networks used to approximate $f(\cdot)$ and $g(\cdot)$ (see section "Methods" for more details).

**Table 1 | The chemical components and basic experiment conditions for four datasets**

| Dataset | Batch | Chemical component | Nominal capacity (mAh) | Cut-off voltage (V) | Experiment temperature (°C) | Number of cells |
|---|---|---|---|---|---|---|
| XJTU | 1,2,3,4,5,6 | $LiNi_{0.5}Co_{0.2}Mn_{0.3}O_2$ | 2000 | 2.5–4.2 | Room temperature | 55 |
| TJU | 1 | $Li_{0.86}Ni_{0.86}Co_{0.11}Al_{0.03}O_2$ | 3500 | 2.65–4.2 | 25,35,45 | 66 |
| | 2 | $Li_{0.84}Ni_{0.83}Co_{0.11}Mn_{0.07}O_2$ | 3500 | 2.5–4.2 | 25,35,45 | 55 |
| | 3 | Blend of 42 (3) wt.% $LiNiCoMnO_2$ and 58 (3) wt.% $LiNiCoAlO_2$ | 2500 | 2.5–4.2 | 25 | 9 |
| MIT | – | $LiFePO_4$ | 1100 | 2.0–3.6 | 30 | 125 |
| HUST | – | $LiFePO_4$ | 1100 | 2.0–3.6 | 30 | 77 |

The charge/discharge protocol varies among different datasets.

vector, and $u^i$ denotes the SOH of the cycle $i$. The nonlinear function $g(\cdot)$ models the SOH decay rate of the battery. Since $f(\cdot)$ and $g(\cdot)$ are affected by many factors in reality and their explicit expressions are unknown, they are replaced by small fully connected neural networks, denote as $\mathcal{F}(\cdot)$ and $\mathcal{G}(\cdot)$. During training, we consider data term loss $\mathcal{L}_{data}$, monotonicity loss $\mathcal{L}_{mono}$, and loss $\mathcal{L}_{PDE}$ constrained by the degradation equation described by partial differential equation. They minimize the errors between the predicted and the true values, while making the model follow the properties of monotonicity of the degradation trajectory and satisfy the constraints of the established degradation model.

To validate the superiority of the proposed PINN, we conducted small sample experiments and transfer experiments. During the transfer experiments, we froze $\mathcal{G}(\cdot)$ and fine-tuned $\mathcal{F}(\cdot)$ on datasets with different chemical compositions. The experimental outcomes demonstrate that the proposed PINN framework can effectively capture the dynamics of battery degradation. Our study combines knowledge of the battery degradation with neural networks and achieves promising results. This study highlights the promise of physics-informed neural network for battery degradation modeling and SOH estimation (more methodological details can be found in the "Methods" section).

### Data generation

To cover different battery types and chemistries, we employ 310,705 samples of 387 batteries from 4 different large-scale datasets for validation. The first dataset comes from the battery degradation experiments we conducted for this study, the other three datasets are well-known public datasets from Zhu et al.[44], Ye et al.[45], and Severson et al.[25]. For convenience, we refer to the four datasets as the XJTU battery dataset, TJU dataset[44], HUST dataset[45], and MIT dataset[25]. The basic information of the four datasets is given in Table 1.

We developed a battery degradation experiment in this study, as shown in Fig. S1. A total of 55 batteries manufactured by LISHEN ($LiNi_{0.5}Co_{0.2}Mn_{0.3}O_2$, 2000 mAh nominal capacity, and 3.6 V nominal voltage, the cut-off voltages of charging and discharging are 4.2 V and 2.5 V) were cycled to failure under 6 charge/discharge protocols at the room temperature. The protocols include fixed charging and discharging, random discharging with a fixed current in different cycles, random walking, and the charging and discharging strategy of a satellite in geosynchronous earth orbit (GEO). We use batch 1 to batch

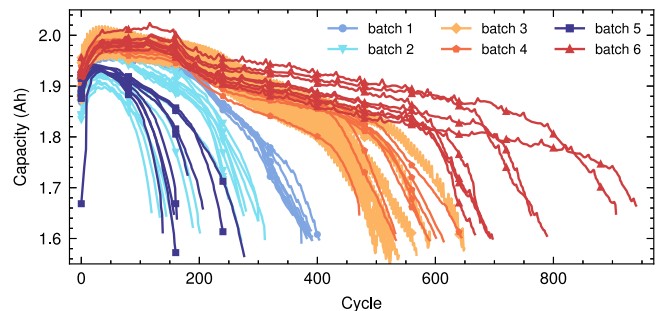

**Fig. 2 | The degradation trajectories of the XJTU battery dataset.** There are 6 batches (55 batteries) in total, all batches contain 8 batteries except batch 2 which contains 15 batteries. The charge/discharge protocols are different among batches. See Supplementary Note 1 for more details.

6 to represent the 6 charge/discharge protocols, respectively. The degradation trajectories are shown in Fig. 2. More details about our dataset can be found in Supplementary Note 1.

The TJU dataset contains three types of battery: NCA battery (3500 mAh nominal capacity and 2.65–4.2 V cut-off voltage), NCM battery (3500 mAh nominal capacity and 2.5–4.2 V cut-off voltages), and NCM + NCA battery (2500 mAh nominal capacity and 2.5–4.2 V cut-off voltage). These batteries are cycled in a temperature-controlled chamber with different temperatures and different charge current rates. Candidate sets for temperatures include 25, 35, and 45 °C. Current rates ranging from 0.25 C to 4 C were used. We use batch 1, batch 2, and batch 3 to represent NCA, NCM, and NCM + NCA batteries, respectively.

The HUST dataset contains data from 77 LFP/graphite cells under 77 different multi-stage discharge protocols. These batteries, manufactured by A123 (APR18650M1A), have a nominal capacity of 1100 mAh and a nominal voltage of 3.3 V. They were cycled at a temperature of 30 °C with an identical charge protocol but different discharge protocols until failure.

The batteries in the MIT dataset have the same type and chemical composition as the batteries in the HUST dataset. However, unlike the experimental setup at the HUST dataset, the MIT dataset considered multiple fast-charging strategies and one discharging strategy.

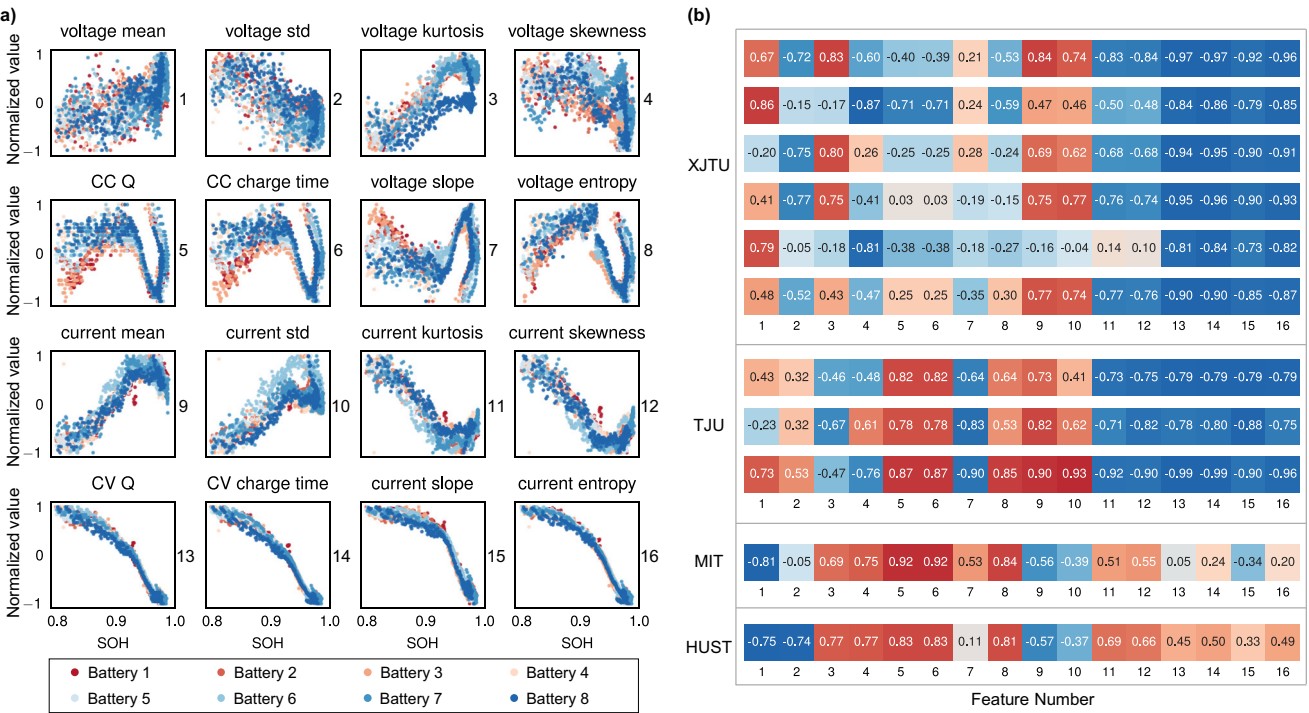

**Fig. 3 | An illustration of extracted features and correlation coefficients.**
**a** Features of 8 batteries from the XJTU dataset batch 1. The *x*-axis of each subfigure is SOH, and the *y*-axis is the normalized value of the corresponding feature. The number on the right side of each subfigure represents the feature number. **b** Correlation heatmap between extracted features and SOH in four datasets. The numbers 1–16 represent 16 features, and the order of features is consistent with that in (**a**).

Specifically, they were cycled under a fast-charging experiment with a one-step or two-step fast-charging policy, and discharged at 4 C. The experiment temperature is 30 °C.

**Feature extraction**

Robust features can often improve the performance of SOH estimation. However, how to extract general and robust features is a worthy research problem. In existing studies, various feature extraction methods for specific datasets and charge/discharge protocols were proposed, yet the generalization of features has been insufficiently considered. There are few studies on methods for extracting general features for different battery types or charge/discharge protocols. To extract more robust and generalizable features, we propose a method to extract features from a short period of charging voltage curve and current curve through observation and exploration of multiple datasets. It is an undoubted fact that the discharging process of the battery is user-specific. In contrast, the charging process is essential and more fixed and regular, and the probability of the battery being fully charged is relatively high. We found that most datasets contain constant current and constant voltage (CC-CV) charging modes. For the four public datasets we used, no matter what strategy the battery is discharged with or whether it is fully discharged, it will eventually be fully charged when charging.

Therefore, we selected a short period of data before the battery was fully charged to extract features, as shown in Fig. 1b. Define the charge cut-off voltage of a battery as $V_{end}$, and the voltage data whose value is within $[V_{end-0.2}, V_{end}]$ V is selected. For the current data, we choose the data with the current between 0.5 A and 0.1 A during the constant voltage charging. Regardless of whether the battery is fully discharged, as long as the battery is fully charged, the voltage range and current range always exist.

The mean, standard deviation, kurtosis, skewness, charging time, accumulated charge, curve slope, and curve entropy from the selected current and voltage curves, respectively (these features are numbered

1–16, respectively. See Supplementary Note 2 for more details) are extracted. An illustration of extracted features from XJTU dataset batch 1 is given in Fig. 3a. Further, we extracted features from 387 batteries in 4 datasets respectively, and calculated the Pearson correlation coefficient between features and SOH within each dataset, as shown in Fig. 3b.

Based on experimental phenomena and analysis of Fig. 3b, we give a natural conjecture: the magnitude of the correlation coefficient between each feature and SOH is related to the chemical composition of a battery and is less affected by the charge/discharge protocols. To the best knowledge, we are the first to focus on this phenomenon. It can be seen from Table 1 that both the XJTU dataset and the TJU dataset are LiNiCo-x type batteries. Even though they have completely different nominal capacities and charge/discharge protocols, the features extracted from our selected range are highly similar. For example, there is a very strong negative correlation between features 11–16 and SOH. Features 9 and 10 have a strong positive correlation with SOH. In contrast, the MIT dataset and the HUST dataset are both LiFePO$_4$ batteries. Features 11–16 show a weak positive correlation with SOH, while features 9 and 10 show a negative correlation with SOH. Besides, features 3–6 and 8 of the latter two datasets show a strong positive correlation with SOH.

**SOH estimation**

The extracted 16 features and time (cycle) are used as inputs of the proposed PINN to estimate SOH. To reduce the impact of the difference in feature magnitude on the model and make the model training more stable, the min–max normalization is performed on the features. That is, all features are scaled to the range [−1,1]. The SOH estimation results of the proposed PINN on 4 datasets are given in Fig. 4a (the number of test batteries in each dataset can be found in Table S2.

To demonstrate the advancement of the proposed PINN, Multi-Layer Perceptron (MLP) with the same structure and parameter amounts and Convolutional Neural Network (CNN) with similar

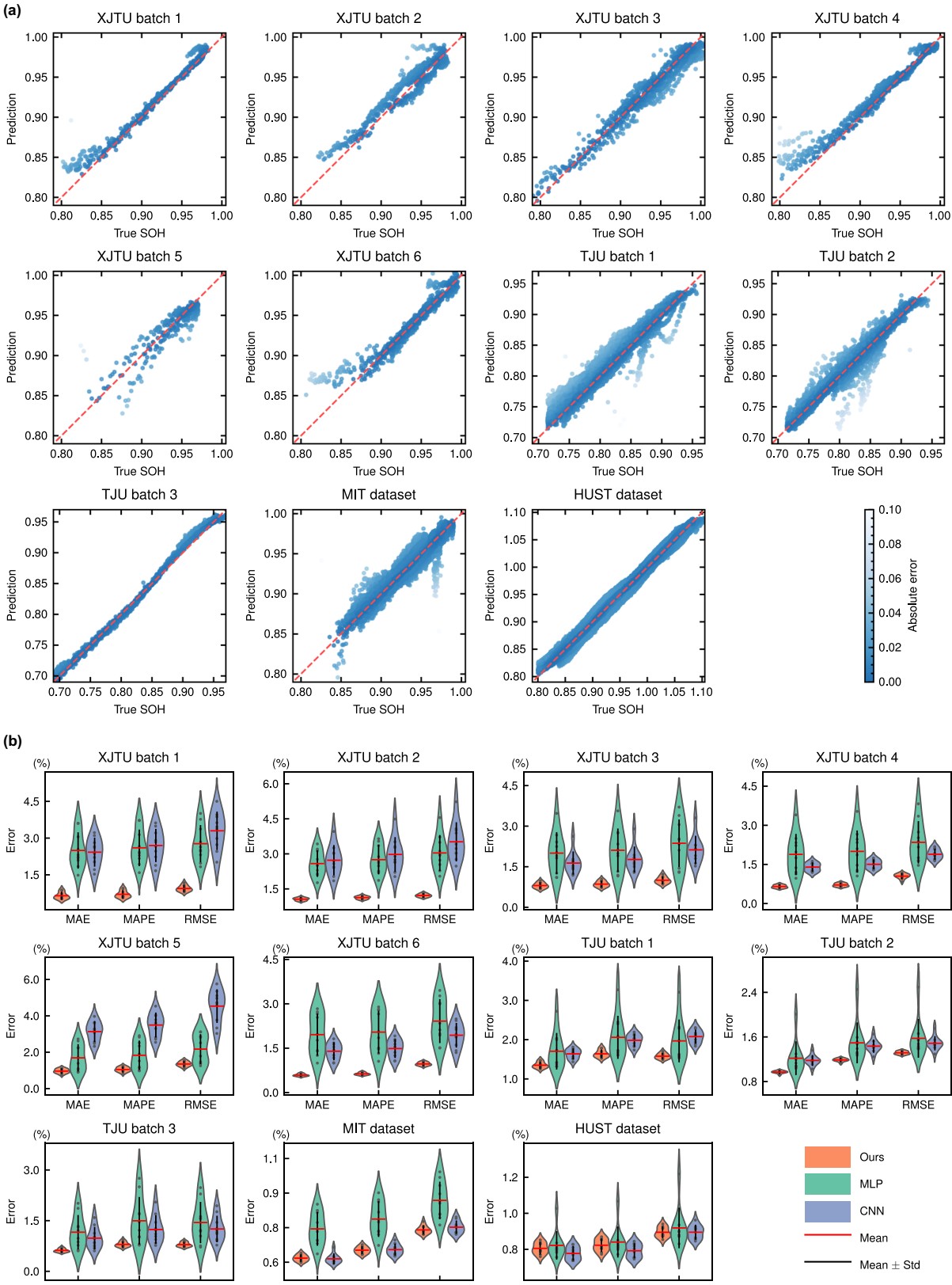

**Fig. 4 | The illustrations of SOH estimation results. a** The SOH estimation results of proposed PINN on four datasets. The predicted and true SOH are distributed near the diagonal, indicating that the model performs well. **b** Distributions of mean absolute error (MAE), mean absolute percentage error (MAPE), and root mean square error (RMSE) of 3 models (the proposed PINN (Ours), multi-layer perceptron (MLP), and convolutional neural network (CNN)) on four datasets. Each error bar contains 10 points (10 experiment) and is marked with mean and standard deviation lines. Compared with the other two methods, our method has smaller prediction errors and is more stable. Source data are provided as a Source Data file.

**Table 2 | The results of proposed PINN (Ours), multi-layer perceptron (MLP), and convolutional neural network (CNN) on four datasets**

| Dataset | Batch | Ours | | MLP | | CNN | |
|---|---|---|---|---|---|---|---|
| | | MAPE | RMSE | MAPE | RMSE | MAPE | RMSE |
| XJTU | 1 | **0.0070** | **0.0094** | 0.0260 | 0.0277 | 0.0270 | 0.0330 |
| | 2 | **0.0113** | **0.0122** | 0.0275 | 0.0304 | 0.0298 | 0.0352 |
| | 3 | **0.0086** | **0.0100** | 0.0211 | 0.0237 | 0.0177 | 0.0212 |
| | 4 | **0.0071** | **0.0105** | 0.0200 | 0.0235 | 0.0150 | 0.0189 |
| | 5 | **0.0105** | **0.0135** | 0.0183 | 0.0217 | 0.0350 | 0.0453 |
| | 6 | **0.0063** | **0.0097** | 0.0204 | 0.0242 | 0.0149 | 0.0194 |
| TJU | 1 | **0.0164** | **0.0158** | 0.0206 | 0.0197 | 0.0198 | 0.0208 |
| | 2 | **0.0119** | **0.0132** | 0.0149 | 0.0157 | 0.0143 | 0.0149 |
| | 3 | **0.0080** | **0.0079** | 0.0150 | 0.0144 | 0.0124 | 0.0125 |
| MIT | | **0.0065** | **0.0074** | 0.0079 | 0.0087 | **0.0065** | 0.0075 |
| HUST | | 0.0078 | **0.0087** | 0.0080 | 0.0090 | **0.0074** | 0.0087 |

MAPE is the mean absolute percentage error, and RMSE is the root mean square error. The best results are shown in bold. All values are averaged from ten experiments.

**Table 3 | Results of small sample experiments on the XJTU dataset batch 1 and HUST dataset**

| Dataset | Train batteries | Ours | | MLP | | CNN | |
|---|---|---|---|---|---|---|---|
| | | MAPE | RMSE | MAPE | RMSE | MAPE | RMSE |
| XJTU | 1 | **0.0141** | **0.0184** | 0.0343 | 0.0390 | 0.0929 | 0.0949 |
| | 2 | **0.0105** | **0.0134** | 0.0267 | 0.0304 | 0.0728 | 0.0826 |
| | 3 | **0.0069** | **0.0096** | 0.0347 | 0.0383 | 0.0548 | 0.0666 |
| | 4 | **0.0056** | **0.0076** | 0.0292 | 0.0327 | 0.0560 | 0.0647 |
| HUST | 1 | **0.0446** | **0.0485** | 0.0601 | 0.0682 | 0.3614 | 0.1550 |
| | 2 | **0.0178** | **0.0202** | 0.0391 | 0.0461 | 0.0826 | 0.0925 |
| | 3 | **0.0154** | **0.0181** | 0.0251 | 0.0287 | 0.0514 | 0.0618 |
| | 4 | **0.0144** | **0.0173** | 0.0253 | 0.0288 | 0.0429 | 0.0521 |

MAPE is the mean absolute percentage error, and RMSE is the root mean square error. The best results are shown in bold. All values are averaged from ten experiments. The "Train Batteries" means that we use 1, 2, 3, and 4 batteries to train the model respectively, and then test it on test set.

parameter amounts are used as comparison methods. The details of MLP and CNN can be found in Supplementary Note 3. For each dataset, we divide the training batteries, validation batteries, and test batteries approximately in a ratio of 6:2:2. The number of test batteries in each dataset can be found in Table S2. To ensure fairness, the numbers of test batteries are evenly distributed throughout the dataset, as shown in Tables S3–S6. The results of the 3 models on the 4 datasets are shown in Table 2 (only the average test errors on each dataset are given, and the test results of each battery can be viewed in Tables S3–S6. It can be seen from the table that our method has the smallest estimation errors in most cases. The average MAPE of the proposed PINN on the 4 datasets is 0.85%, 1.21%, 0.65%, and 0.78%, while that of MLP is 2.60%, 1.72%, 0.83%, and 0.83%. It is worth noting that they have the same number of parameters and model structure during inference.

Further, to reflect the stability of the model, the training and testing process of each model on each dataset is repeated 10 times. The test results are shown in Fig. 4b. From the figure, we can see that our proposed PINN is the most stable on all tasks and all metrics. The sample size in each batch of the XJTU battery dataset is small, causing significant fluctuations in MLP and CNN. In contrast, our method is more stable and yields a smaller test error. For HUST dataset and MIT dataset, they contain a large number of training samples, so the fluctuations of MLP and CNN become smaller, and the test errors become smaller. However, our proposed PINN is still the best-performing model.

### Experiments with small samples
Our proposed PINN models battery degradation dynamics, taking into account more physical laws and thus can be trained with less data. Compared with pure data-driven methods, our method can show greater superiority when the amount of available training data is small. To verify the above inference, small sample experiments are conducted on the XJTU dataset and HUST dataset.

Specifically, we use 1 battery data to train 3 models, and test on multiple batteries (the test set is the same as in 2.4), and record the test results. In addition, we gradually increase the number of training batteries and observe the performance change of each model on the test set. The results are given in Table 3 and Fig. 5 (only the batch 1 results are given for the XJTU dataset, more results can be found in Table S7 and Fig. S5).

It can be observed that our proposed PINN obtains the best results in all tasks and settings. As the number of training batteries increases,

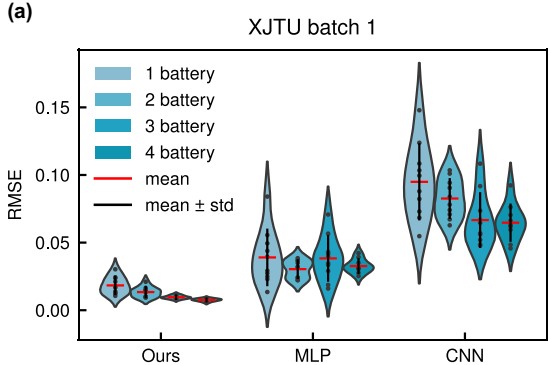

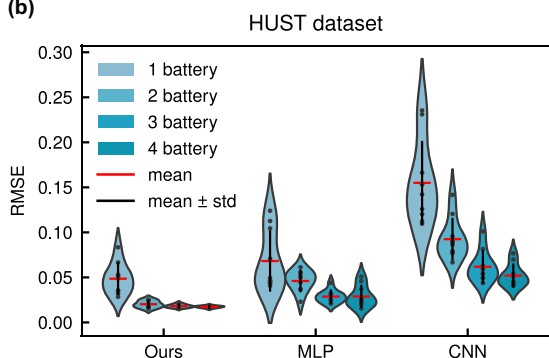

**Fig. 5 | An illustration of test root mean square error (RMSE) distributions for three models (the proposed PINN (Ours), multi-layer perceptron (MLP), and convolutional neural network (CNN)) on two datasets.** Each error bar contains 10 points (10 experiment) and is marked with mean and standard deviation lines. The "1 battery" in the legend means that we only use the data of 1 battery to train the model. Others are similar. As the number of batteries increases, the performance of the three models is getting better. However, our method still performs best among them. **a** The results on the XJTU dataset batch 1. **b** The results on the HUST dataset. Source data are provided as a Source Data file.

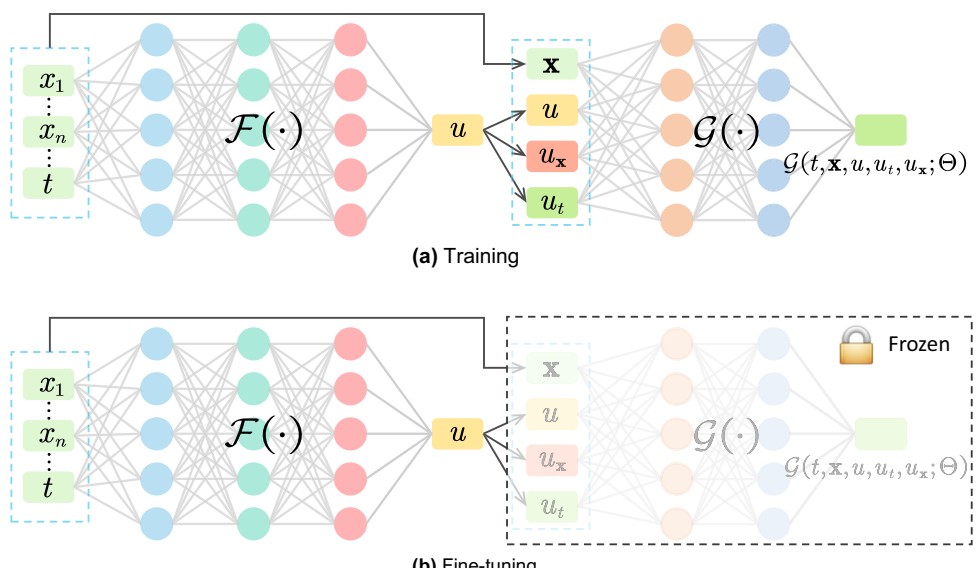

**(a)** Training

**(b)** Fine-tuning

**Fig. 6 | An illustration of the proposed physics-informed neural network. a** The extracted features **x** and cycle $t$ are used to estimate SOH $u$. The $\hat{u}$ represent the estimated SOH, and the subscripts $t$ and **x** represent the corresponding partial derivatives. Neural networks $\mathcal{F}(\cdot)$ and $\mathcal{G}(\cdot)$ is used to model the mapping between features to SOH and the degradation dynamics of battery, respectively. **b** When the proposed PINN is applied to transfer learning scenarios, the dynamics $\mathcal{G}(\cdot)$ is frozen and only solution $\mathcal{F}(\cdot)$ is fine-tuned.

**Table 4 | The test root mean square error (RMSE) of fine-tuning experiments among four datasets**

| | Fine-tuning | | | | Source-only | | | | Train with target cell | |
|---|---|---|---|---|---|---|---|---|---|---|
| | XJTU | TJU | MIT | HUST | XJTU | TJU | MIT | HUST | 1 | 2 |
| XJTU | – | **0.0100** | 0.0145 | *0.0104* | – | 0.0967 | 0.1329 | 0.0733 | 0.0184 | <u>0.0134</u> |
| TJU | **0.0093** | – | *0.0119* | 0.0146 | 0.1266 | – | 0.1674 | 0.1266 | <u>0.0121</u> | 0.0202 |
| MIT | *0.0239* | 0.0272 | – | <u>0.0248</u> | 0.0347 | 0.1277 | – | 0.0561 | 0.0324 | **0.0142** |
| HUST | <u>0.0333</u> | 0.0343 | *0.0307* | – | 0.1131 | 0.2008 | 0.0801 | – | 0.0485 | **0.0202** |

The top 3 results are in bold, italic, and underlined respectively. All values are averaged from ten experiments. View the table in terms of rows. The first row represents that the XJTU dataset is used as the target domain, and other datasets are used as the source domain. "Fine-tuning" means that the PINN was trained on the source domain, then fine-tuned with the data from the 1st battery in the target domain, and tested on the test set of the target domain. "Source-only" means that the PINN was trained on the source domain, and then tested on the test set of target domain directly. "Train with target cell" represents that the PINN was trained with 1 or 2 batteries from the target domain and then tested on the test set of the target domain (the same as the small sample experiments). For convenience, we only select the data in XJTU batch 1 to represent the XJTU dataset. Similarly, batch 3 of the TJU dataset is used to represent the TJU dataset.

the test errors decrease for all 3 models. This is a generally accepted fact: increasing the number of training samples can improve the model performance when the training data is small. In Fig. 4b, due to the large number of samples in the HUST dataset, the performance of the CNN and MLP is comparable to that of our PINN. This also illustrates the fact that when the structure or number of parameters of the models are the same or similar, given enough training samples, the model performance does not differ much. However, it is evident from Fig. 5 that our method has a significant advantage when the number of training samples is small. In addition, it is worth noting that the performance of our PINN trained with only 1 battery is comparable to that of MLP and CNN trained with 3–4 batteries, which demonstrates the superiority of our PINN in the small sample scenario.

**Fine-tuning between different datasets**

Fine-tuning is one implementation of transfer learning, which improves learning ability by rapidly tuning the model using a small amount of newly collected data. The advantage is that it can use the massive data collected in other scenarios (source domain) to pre-train a model and learn the essential relation between features and labels. Then a small amount of target domain data is used to quickly fine-tune the model to obtain good performance. Most of the existing studies on transfer learning for SOH estimation are transfers between different charge/discharge protocols, and there are few studies on transfers between different datasets (different chemical compositions). In this paper, we combined 4 datasets in pairs for fine-tuning experiments.

We believe that the degradation dynamics $\mathcal{G}(\cdot)$ are independent of charge/discharge protocols and datasets, while the solution $\mathcal{F}(\cdot)$ is related to them. After learning from massive data, $\mathcal{G}(\cdot)$ should contain general information that can reflect the nature of battery degradation, which is useful for cross-scenario SOH estimation. Therefore, we only fine-tune the weights of the solution $\mathcal{F}(\cdot)$ and make the weights of dynamics $\mathcal{G}(\cdot)$ frozen, as shown in Fig. 6b.

We carried out fine-tuning experiments and source-only experiments, and also compared them with the small sample experiments. All results are given in Table 4. It can be seen from the figure that the fine-tuned model is significantly better than the source-only method. What is more, when there is only 1 labeled target domain battery, the models following the "pre-training−fine-tuning" paradigm perform better than models trained directly using 1 target domain battery. This demonstrates the effectiveness of the "pre-training−fine-tuning" paradigm. For the XJTU dataset and TJU dataset, even if the model is trained with 2 target domain batteries, its performance is not as good as that of the model fine-tuned with 1 target domain battery. This also proves that dynamics $\mathcal{G}(\cdot)$ has learned useful information from a large amount of data in the source domain.

There also seem to be some intuitively correct but less obvious insights if Table 4 is revisited from a fairer perspective, i.e., ignoring

the last column of the table. Both the XJTU dataset and the TJU dataset are LiNiCo-x batteries, and the correlation between the features and SOH is more similar (see section "Feature extraction" for correlations), so the fine-tuning effect between them is better. Similarly, both MIT and HUST are LiFePO$_4$ batteries, and the fine-tuning effect between them is also promising. This may be a meaningful finding, and we will continue to study it in the future.

## Discussion

Accurate SOH estimation facilitates health management and maintenance decisions of lithium-ion batteries. Existing SOH estimation methods need to extract different features for different datasets, and the performance of the model fluctuates greatly. In this work, we propose a general PINN for battery SOH estimation. Specifically, we propose a general feature extraction method to extract statistical features from a short period of data before the battery is fully charged, which is included in batteries charged with a constant-current and constant-voltage mode. Then, we modeled the battery degradation dynamics with a PINN, and the SOH was estimated by taking the extracted features as inputs.

To validate our approach, we performed battery aging experiments and developed a dataset with 55 batteries. Finally, we validate our method on 387 batteries with different chemistries and charge/discharge protocols from 4 large-scale datasets. The results demonstrated the effectiveness and feasibility of our proposed method. Further, we conduct small sample experiments and transfer experiments, proving that considering physical knowledge helps data-driven models to learn faster and better from data. Our study highlights the promise of physics-informed machine learning in battery degradation modeling and SOH estimation. It can facilitate the rapid development of battery management systems for next-generation batteries using existing experimental data and small new data.

Battery degradation modeling and SOH estimation are research hotspots in the field of battery health management. As batteries aging, various interface degradation processes occur, along with the loss of lithium inventory and active materials, leading to increased resistance in ion and electron transfer as well as intercalation reactions, thereby resulting in changes in their charging curves[46]. Consequently, the charging curve contains rich information on the degradation process. However, using charge and discharge curves to estimate battery SOH may fall into the trap of information leakage. Geslin et al.[47] pointed out that inconsistent charging and discharging protocols, usage conditions, etc. may lead to information leakage, which is a serious problem that may be ignored by scholars. They believe that a fixed CC-CV mode can alleviate the problem of information leakage. Hence, it is advisable to avoid incorporating factors related to internal battery quality, manufacturing variability, and usage conditions as much as possible when performing SOH estimation tasks. In our study, the features are extracted from a small segment of data from the CC-CV stage before the battery is fully charged, which is independent of the battery usage conditions. This ensures the usefulness and versatility of the features we extracted, while avoiding the problem of information leakage caused by inconsistent charging protocols or battery usage conditions. During the training and test stage, we train the model with data from battery A and test it on battery B; instead of training the model with early data from battery A and testing it with later data, which avoids information leakage from the training set to the test set.

When building the SOH estimation model, we proposed a PINN for battery SOH estimation. Physics-informed neural network holds promise as an effective avenue for leveraging artificial intelligence to address practical engineering problems. By amalgamating traditional physics models with neural networks models, it can more accurately capture the intricate dynamic behavior of battery systems, thereby facilitating more reliable and precise state estimation. However, this burgeoning field still requires further exploration by scholars. Within

the framework proposed by Aykol et al.[38], Hybrid methods, which utilize physical equations to constrain neural networks or integrate physical equations into neural networks, will become dominant in the long term. This class of hybrid methods have the potential to blend the causality and extrapolation capabilities of physics-based models with the speed, flexibility, and high-dimensional capabilities of neural networks. However, the limitation of these methods lies in the complexity of the battery's physical model (e.g., the P2D model), which contains numerous parameters, and the internal parameters of the battery are difficult to collect. There is currently no satisfactory method to seamlessly integrate physical models and neural networks. The PINN proposed in this paper is modeled from the perspective of empirical degradation and state space equations, serving merely as an exploration of such hybrid methods and acting as a catalyst for further research. Additionally, we only consider extracting features from easily accessible current and voltage data. As more data and internal variables become available, more complex electrochemical models can be considered. The optimal integration of battery governing equations and neural networks for health management within the constraints of existing data and computational resources remains ripe for further exploration.

## Methods

### Battery degradation modeling

Battery aging is primarily characterized by a decrease in available capacity and an increase in internal resistance, typically following a declining trajectory. To accurately describe the battery degradation trajectory, scholars have proposed various empirical models to describe the loss of battery capacity as a function of time or cycle numbers, including the linear model[48], exponential model[49,50], power-law model[51], and failure forecast model (FFM)[52], etc. These models all describe the battery's degradation trajectory as a univariate function of time.

However, representing the degradation trajectory of batteries solely as a univariate function of time oversimplifies the process. In fact, battery degradation is not only related to time but also related to charging rate, discharging rate, calendar time, temperature, depth of discharge (DOD), etc. For example, Xu et al.[53] divided battery aging into calendar aging and cycle aging, which considered factors such as state-of-charge (SOC), DOD, cell temperature, and solid electrolyte interphase (SEI) film growth. They modeled calendar aging and cycle aging as functions of calendar time, SOC, DOD, and temperature.

Therefore, modeling the degradation trajectory of a battery solely as a function of time is inadequate. In this study, we propose to model it as a multivariate function:

$$u = f(t, \mathbf{x}), \tag{1}$$

where $t$ represents time and $\mathbf{x}$ represents a vector composed of SOC, DOD, temperature, charge rate, discharge rate, health indicators (HIs), and all other factors. In our work, $\mathbf{x}$ represents the HIs extracted from the charging data (see "Feature extraction" section for more details).

Without loss of generality, to describe the degradation dynamics of the battery, its SOH decay rate can be described as:

$$\frac{\partial u}{\partial t} = g(t, \mathbf{x}, u; \theta). \tag{2}$$

The above equation is an explicit partial differential equation (PDE) parameterized by $\theta$, and $g(\cdot)$ represents the nonlinear function of $t$, $\mathbf{x}$, and $u$. The function $g(\cdot)$ characterizes the internal degradation dynamics of the battery, and by altering this nonlinear function, various forms of degradation can be represented. Models such as linear model, exponential model, power-law model, and FFM can be viewed as particular cases of Eq. (2) when only the time is considered.

## Physics-informed neural network

An unavoidable problem is that the explicit form of $g(\cdot)$ is unknown and difficult to obtain. In response to similar problems, Sun et al.[36] proposed a sparse regression physics-informed neural network that exploits sparsity to learn the parameters $\theta$ of $g(\cdot)$ from a given candidate set. Raissi et al.[54] proposed deep hidden physics models to model $g(\cdot)$. Inspired by[54,55], we propose to use a more generalized function approximator $g'(\cdot)$ with parameters $\theta'$ to represent the nonlinear dynamics $g(\cdot)$. Therefore, Eq. (2) becomes:

$$u_t \approx g'\left(t, \mathbf{x}, u, u_t, u_{\mathbf{x}}, u_{\mathbf{xx}}, \cdots ; \theta'\right). \tag{3}$$

In the equation, $u_t = \frac{\partial u}{\partial t}$, we employ a neural network $\mathcal{F}(t,\mathbf{x};\Phi)$ with learnable parameters $\Phi$ to model $f(t,\mathbf{x})$ and utilize automatic differentiation mechanisms to compute $u_t$. $u_{\mathbf{x}} = \left[\frac{\partial u}{\partial x_1}, \frac{\partial u}{\partial x_2}, \cdots\right]^{\top}$ represents the first-order partial derivative of $u$ with respect to $\mathbf{x}$, and $u_{\mathbf{xx}}$ represents the second-order partial derivative. One advantage of this approach is that we do not need to specify a candidate basis function set for $g(\cdot)$, but instead employ a more generalized approximators $g'(\cdot)$. The function approximator $g'(\cdot)$ propose a more flexible relationship to $t, u$, $\mathbf{x}$, and their arbitrary order partial derivatives. A neural network $\mathcal{G}(\cdot)$ with learnable parameters $\Theta$ is used to model $g'(\cdot)$ so that it can learn the aging mechanism of the battery from the given $\mathbf{x}, t$, and other partial derivatives. To balance accuracy and computational complexity, we only consider the influence of first-order partial derivatives, discarding higher-order derivatives.

Building upon the aforementioned analysis, we define a physics-informed neural network $\mathcal{H}$[37,55] for battery aging:

$$\mathcal{H} := \frac{\partial \mathcal{F}(t,\mathbf{x};\Phi)}{\partial t} - \mathcal{G}\left(t, \mathbf{x}, u, u_t, u_{\mathbf{x}}; \Theta\right), \tag{4}$$

where $\frac{\partial \mathcal{F}(t,\mathbf{x};\Phi)}{\partial t}$ represents the partial derivation of solution neural network $\mathcal{F}(\cdot)$ with respect to $t$, and $\mathcal{G}(\cdot)$ denotes the battery degradation dynamic equation modeled by the neural network. The structure of the proposed PINN is shown in Fig. 6.

Equation (4) is derived from Eqs. (2) and (3). However, since it is fitted by a neural network, its training process is discrete, so it does not strictly satisfy Eq. (2). For battery SOH, the calculation formula is[13]:

$$u^k = f(k, \mathbf{x}) = \frac{Q^k}{Q^0}, \tag{5}$$

where $Q^k$ represents the capacity of cycle $k$ and $Q^0$ represents the nominal capacity. The SOH value $u^k$ coincides with the point on the degradation trajectory $f(\cdot)$ when $t = k$. We need to make $\mathcal{H}(t^i, \mathbf{x}^i) = 0$ hold at sample point $i$ to approximate Eq. (2). Therefore, the optimization process of the PINN needs to adhere to the PDE loss specified by Eq. (2), i.e.:

$$\mathcal{L}_{\mathrm{PDE}} = \sum_{i=1}^{N} \left|\mathcal{H}\left(t^i, \mathbf{x}^i\right)\right|^2, \tag{6}$$

where superscript $i$ denotes the $i$th sample and $N$ denotes the number of samples. Also, the optimization objective includes data item loss and monotonicity loss:

$$\mathcal{L}_{\mathrm{data}} = \sum_{i=1}^{N} \left|u^i - \hat{u}^i\right|^2, \tag{7}$$

$$\mathcal{L}_{\mathrm{mono}} = \sum_{j=1}^{M} \sum_{k=1}^{N_j} \mathrm{ReLU}\left(\hat{u}^{k+1} - \hat{u}^k\right), \tag{8}$$

where $\hat{u}^i$ represents the estimated SOH, $M$ represents the number of batteries, $N_j$ denotes the number of cycles of battery $j$, and $\mathrm{ReLU}(\cdot)$ is Rectified Linear Unit. The monotonicity loss $\mathcal{L}_{\mathrm{mono}}$ is based on the physical properties of battery degradation, that is, the SOH of the next cycle should be less than or equal to that of the previous cycle (unless capacity regeneration occurs). The total function is formulated as:

$$\mathcal{L} = \mathcal{L}_{\mathrm{data}} + \alpha \mathcal{L}_{\mathrm{PDE}} + \beta \mathcal{L}_{\mathrm{mono}}, \tag{9}$$

where the $\alpha$ and $\beta$ are trade-off parameters. More details about our model can be found in Supplementary Note 3.

## Transfer learning with physics-informed neural network

Our PINN for battery aging consists of two parts: a solution neural network $\mathcal{F}(\cdot)$ that builds the feature-to-SOH mapping and a neural network $\mathcal{G}(\cdot)$ that models battery degradation dynamics, as shown in Fig. 6. We believe that the degradation dynamics $\mathcal{G}(\cdot)$ are independent of charge/discharge protocols and datasets, while the solution $\mathcal{F}(\cdot)$ is related to them. Therefore, when our PINN is applied to transfer learning scenarios, dynamics $\mathcal{G}(\cdot)$ is frozen, and only solution $\mathcal{F}(\cdot)$ is fine-tuned, as shown in Fig. 6b.

## Data availability

The XJTU battery dataset generated in this study is publicly available in the Zenodo database under accession code [https://doi.org/10.5281/zenodo.10963339], as reference[56]. The TJU dataset is available at: https://zenodo.org/record/6405084. The HUST dataset is available at: https://data.mendeley.com/datasets/nsc7hnsg4s/2. The MIT dataset is available at: https://data.matr.io/1/projects/5c48dd2bc625d700019f3204. Source data are provided with this paper.

## Code availability

Our code is available on Github [https://github.com/wang-fujin/PINN4SOH] or on Zenodo database under accession code [https://doi.org/10.5281/zenodo.11046967], as reference[57].

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

## Acknowledgements

This work was supported in part by the National Natural Science Foundation of China under Grand 52105116 (Z.Z. (Zhibin Zhao)) and Grand 92060302 (X.C.); the Fundamental Research Funds for the Central Universities (xzy022023060) (F.W.).

## Author contributions

F.W. was responsible for conceptualization, methodology design, conducting experiments, and drafting the original manuscript. Z.Z. (Zhi Zhai) extensively reviewed and edited the manuscript, providing valuable suggestions and revisions. Z.Z. (Zhibin Zhao) contributed to conceptualization and methodology discussions, playing a significant role in the editing process. Y.D. conducted comprehensive reviews and edits, significantly contributing to the refinement of the article. X.C. spearheaded the acquisition of funds necessary for this research, providing crucial support.

## Competing interests

The authors declare no competing interests.
