## [Peer Review File · Nature Communications]

REVIEWER COMMENTS

Reviewer #1 (Remarks to the Author):

The authors should make an effort to appropriately position their research, as they mix different concepts and techniques without fully referencing the background of their contribution. For instance, on line 12 of page 3, they state that there are two challenges in SOH estimation: "the generality of degradation features and the stability of data-driven models." This claim is quite broad and should either be justified or supported by a citation. The given explanation of both challenges is not persuasive, thus not adequately justifying the use of PINN for SOH modeling in that paragraph; they only include general references about the technique and don't cite recent works on using PINN for battery SOH modeling (the latest reference provided by the authors, unless I'm mistaken, is [36] from 2021). For instance, in reference [14] provided by the authors, more recent studies are mentioned, such as:

Nascimento, R.G., Viana, F.A.C., Corbetta, M. et al. A framework for Li-ion battery prognosis based on hybrid Bayesian physics-informed neural networks. *Sci Rep* 13, 13856 (2023).
<https://doi.org/10.1038/s41598-023-33018-0>

It's challenging to pinpoint the exact contribution of the paper: it isn't the first to use PINNs for battery SOH modeling, and the SOH model used is simpler than others previously employed. They also don't specify which of the open topics in SOH estimation mentioned in [14] are addressed (e.g., transferring the model across different chemistries and charge profiles or capacity recovery after resting). The introduction needs revision, and the authors should clearly state their paper's unique contribution to the field and cite the latest related works.

Section 2 outlines their method implementation. It begins with the network architecture and data generation. In section 2.3, an essential aspect is discussed: the authors have chosen a 0.2 V range between the maximum charge voltage (or minimum discharge), but they don't clearly detail the conditions under which this measurement should occur (I miss specifications on the resting period required after full charge or discharge and its relation to the charging rate). This section needs more details and a better justification. In section 2.4, there's a brief comparison with a convolutional network. This could be enhanced by incorporating SOH estimation methods from other authors in the literature. Currently, they've shown that a neural architecture using physical knowledge outperforms a convolutional one, but this finding isn't very informative. When comparing network errors, it seems they are using features they've defined, essentially contrasting two algorithms they've developed, not benchmarking against the state-of-the-art in SOH data estimation. Moreover, they don't define the function \mathcal{F} incorporating the physical knowledge until section 4, making it necessary to skip forward and then backtrack to understand these results. The document's organization could be improved.

In section 4, the authors introduce a function modeling physical knowledge about SOH's temporal evolution. This section is somewhat challenging to follow. Equation (1) defines a SOH value sequence indexed by cycle. Equation 2 references a continuous variable $u(t)$; I gather u_k is a discretization of $u(t)$. The function \mathcal{F} is the derivative of $u(t)$ and depends on time, health, and a parameter. Equation 5 expands u to depend on a parameter x (though u is already defined as \mathcal{F} 's integral, thus depending on the previously introduced parameter θ). Equation 6 redefines \mathcal{F} to depend on two parameters and also states u as a partial derivative of \mathcal{F} , deviating from equation (2). By equation (7), the explanation becomes muddled: it defines a function subtracting u_t (that was defined in equation (1) as a SOH sequence) from its partial derivative concerning time, indicating that u_t becomes a derivative. Yet, this implies that, per definition (7), $f := 0$. Equation (8) redefines u_t to include u 's gradient concerning the parameter vector x ; by the same logic, I assume u_{xx} is this function's Hessian, though it doesn't seem to be used later. There are other issues of varying significance and other technical quality aspects, but at this point, the authors should revise this section and address these concerns before the manuscript can be considered.

Reviewer #2 (Remarks to the Author):

There are three issues to address.

1) The first, the plot in figure 4 did not properly convert to pdf. I can't see the plot. I tried different pdf viewers and all with the same results.

2) I am not convinced that the good model performance is not simply due to information leakage. <https://www.sciencedirect.com/science/article/abs/pii/S2542435123003197>
The current article doesn't ever really explain what the task of prediction is. Since SOH depends on both the battery internal quality as well as the us conditions, the linked article above argues that it is critical to be careful in defining the task you are trying to actually do (beyond the simple reply of "predict SOH"). What are you really trying to predict? Since the datasets were all created with different hypothesis in mind, it is hard to determine whether leakage is happening by using the same features across the data. The authors should more clearly define their task and clarify how they have ensured leakage is not the reason for good performance. Leakage can fool even careful practitioners.

3) What the physics are that are being included is not really clear from the writing. It kind of exists in section 4, but it is hard to really figure out at a high level what physics are being accounted for. In an early section, a clearer description of the physical model(s) would be helpful.

Reviewer #3 (Remarks to the Author):

In this manuscript, neural networks are used to estimate the SOH from statistical parameters during a period of (full) charge. This first neural network is then used as the input of a second PINN to estimate the progression of the SOH over time. The application to different, comprehensive datasets of cells with different chemistry, make the work interesting, especially the application of transfer learning.

We suggest accepting the manuscript after major revisions, as some work needs to be done to make the manuscript more readable and to provide additional information on parameterization and selection of hyperparameters to make this a valuable contribution to the community. We would recommend adapting the structure of the manuscript so that the method is explained first, then the results are presented and discussed and finally there should be a conclusion section. Special attention should be paid to avoid jumping between the manuscript and supplementary material. Thus, all relevant information, especially on the features, should also be included in the manuscript.

Please consider the following remarks:

- In the abstract, please name the error metric (MAE, RMSE) you refer your final validation results to.

- In the Introduction, please give more details about your chosen PINN architecture with reference to other works, i.e. , Aykol et al. (10.1149/1945-7111/abec55) or Hofmann et al. (10.1149/1945-7111/acf0ef). With the current introduction, it does not get directly clear where the physics-based modelling takes place. It more looks like it's a sequential NN with a novel feature extraction method.

- Please include the paper by Hofmann et al. (10.1149/1945-7111/acf0ef) into your reference list as they have also introduced a sequential PINN for SOH estimation.

- In Figure 2 please provide the units in your axis-labels, i.e., "Capacity / Ah". Please include a legend to allow mapping every batch to the specific degradation curves. Further, make the Figure bw-readable by using different linestyles.

- We understand that the statistical features are all connected to the end of the charge process and this approach is possible for all applications, where a full charge is performed frequently. However, it is not true that for all or even most applications a full charge is performed frequently. In contrary users are directed to stop charging at 80% by special apps to avoid increased aging.

- It is claimed, that "An important finding is that the magnitude of the correlation coefficient between each feature and SOH is related to the chemical composition of a battery and is less affected by the charge/discharge protocols." However, there is no evidence given. Please refrain from that claim or provide evidence.
- Please extend Figure 3 to also show which time-series data is additionally used for SOH estimation. From the text and Figures it gets very clear which data is used for feature extraction, but it is hard to grasp which time-series data is additionally used for SOH estimation together with the extracted features.
- In Figure 5 please make clear which feature number refers to which feature. If the order is consistent with Figure 4, please mention that again. In the text, also include that you have used the Pearson correlation coefficient.
- In figure 7 the y-axis should be scaled equal for all subplots for better comparability of the performance with respect to different datasets. Again, errors given in % are more readable than fractions (i.e. 1% instead 0.01).
- Please comment on the different correlations of the statistical features with respect to chemistry. It is not obvious why there should be a difference between NMC and LFP cells.
- As some features seem to be highly correlated, could you please comment on the selection of the features? How did you decide to incorporate all 16 features?
- In Section 2.4 please include more information about the data split. How did you make sure that batteries from the same batch are not too similar and hence the test datasets is too similar to the training dataset? How much percentage is training and how much is test data?
- Please provide information about the hyperparameter tuning for the proposed PINN. Which architectures were explored and which approach (e.g. Bayes, Random Search, Grid Search) was used for optimizing the hyperparameters.
- Please provide an updated benchmark to create a fair comparison of the PINN to the MLP and CNN. This means, that the hyperparameters of the MLP and CNN should be tuned to their respective optima. In this matter, also include the computational complexity of all three models.
- Please expand the explanations on the PINN: which aspects in this model are to be understood as physical boundary conditions and which are not? In particular, since the partial derivatives of the SOH with respect to the statistical features are included, it is not obvious which physical, chemical boundary conditions are to be considered here.

Reviewer #4 (Remarks to the Author):

I co-reviewed this manuscript with one of the reviewers who provided the listed reports. This is part of the Nature Communications initiative to facilitate training in peer review and to provide appropriate recognition for Early Career Researchers who co-review manuscripts

Response Letter

Journal Title: Nature Communications

Manuscript ID: NCOMMS-23-46244

Title of Paper: Physics-informed neural network for lithium-ion battery degradation stable modeling and prognosis

Note: In this Response Letter, the **contents in blue** are the responses and discussions to the comments of editors and reviewers; the **contents in red** indicate that these contents have been added to the revised manuscript.

First of all, the authors would like to express their sincere thanks to the anonymous reviewers for their helpful comments and suggestions. The explanation of the modifications as well as corrections in this revision can be arranged as follows (comment numbers are in 1:1 correspondence with the reviewers' comments).

1 Response to Reviewer #1

Comment: The authors should make an effort to appropriately position their research, as they mix different concepts and techniques without fully referencing the background of their contribution. For instance, on line 12 of page 3, they state that there are two challenges in SOH estimation: “the generality of degradation features and the stability of data-driven models.” This claim is quite broad and should either be justified or supported by a citation. The given explanation of both challenges is not persuasive, thus not adequately justifying the use of PINN for SOH modeling in that paragraph; they only include general references about the technique and don’t cite recent works on using PINN for battery SOH modeling (the latest reference provided by the authors, unless I’m mistaken, is [36] from 2021). For instance, in reference [14] provided by the authors, more recent studies are mentioned, such as:

Nascimento, R.G., Viana, F.A.C., Corbetta, M. et al. A framework for Li-ion battery prognosis based on hybrid Bayesian physics-informed neural networks. Sci Rep 13, 13856 (2023). <https://doi.org/10.1038/s41598-023-33018-0>

It’s challenging to pinpoint the exact contribution of the paper: it isn’t the first to use PINNs for battery SOH modeling, and the SOH model used is simpler than others previously employed. They also don’t specify which of the open topics in SOH estimation mentioned in [14] are addressed (e.g., transferring the model across different chemistries and charge profiles or capacity recovery after resting). The introduction needs revision, and the authors should clearly state their paper’s unique contribution to the field and cite the latest related works.

RESPONSE: Thank you for your valuable comment. we appreciate the opportunity to clarify and enhance the positioning of our research within the context of the SOH estimation for batteries. Regarding the statement on page 3, line 12, noted by the reviewer, we realize that it is indeed too broad and does not go far enough to elicit the justification for using PINN for SOH modeling. We are grateful to the reviewer for pointing this out. In this revised manuscript, we have modified it to make it more reasonable and naturally lead to the rationality of our modeling using PINN. The specific modifications are as follows:

“ However, challenges still stand in the way of developing reliable, accurate, and general SOH estimation methods [1; 2]. Physics-based models are stable and accurate, but batteries with different chemical compositions require different model parameters, and the models have high computational costs [3]. The data-driven models have high accuracy and efficiency, but its generalizability depends on the extracted features and have poor stability [2; 4]. ...”

“ The promising prospect of physics-informed neural network (PINN) [5; 6] lies in amalgamating the strengths of physics-based and data-driven approaches, potentially addressing the aforementioned challenges. ...”

In the original manuscript, our reference [36], named *Perspective—combining physics and machine learning to predict battery lifetime*, primarily encapsulates methods integrating physics knowledge and machine learning for battery modeling and prognosis. In response to the reviewer’s comment about our paper didn’t cite recent works on using PINN for battery SOH modeling, we have reviewed and cited some related papers in this revised manuscript. The addition is outlined below:

“... It is a promising approach in the field of battery prognosis and diagnostics. Aykol et al. [7] presented 5 approaches that combine physical knowledge and machine learning for battery modeling. Within this framework, some work has been published [8; 9; 10; 11; 12]. Nascimento et al. [8] directly implemented the numerical integration of principle-based governing equations through recurrent neural networks to simulate the dynamic response of the battery. Wang et al. [11] proposed a battery neural network (BattNN) for discharge voltage prediction based on the equivalent circuit model (ECM). Hofmann et al. [12] used the pseudo-two-dimensional (P2D) Newman model to generate data at different health status points and combined it with experimental data and field data to train the neural network model, which takes advantage of the correlation between internal states and measurable SOH. ”

The reviewer pointed out: “It’s challenging to pinpoint the exact contribution of the paper: it isn’t the first to use PINNs for battery SOH modeling, and the SOH model used is simpler than others previously employed.” We make a clear clarification here. There are two types of tasks in the field of battery modeling and health management. One is a task within one charging/discharging cycle, including state-of-charge (SOC) estimation, end-of-discharge (EOD) prediction and other tasks. The other is the task in the entire life cycle, including SOH estimation, remaining-useful-life (RUL) prediction, degradation trajectory prediction. The dynamics of the battery in one cycle can be accurately described by physical models such as electrochemical models (pseudo-two-dimensional (P2D) model, single particle (SP) mode, etc.) and equivalent circuit models (ECM). However, there are only some experience models in the entire life cycle for battery degradation process. Most of the existing PINN methods are for SOC estimation or EOD prediction tasks. For example, the work mentioned by the reviewer: *Nascimento, R.G., et al. A framework for Li-ion battery prognosis based on hybrid Bayesian physics-informed neural networks*, and other papers [13; 14]. These methods can be combined with physical models such as P2D or ECM, thus appear to be more complex than our method.

Of course, there is also a small amount of work using PINN for SOH estimation, such as [9; 10; 15]. According to the classification in reference [36], our method is different from existing methods. The classification given in literature [36] is shown in Figure 1. Most of the existing methods belong to the A1 and A2 category, which mainly use physical models to

pre-process data. Aykol et al. has said in the paper that “*Sequential integration of standalone PB and ML modeling tools (type A architectures) is a well-defined challenge focused primarily on software development and integration. **Sequential architectures are solvable today, through integration, improvement, and repurposing of existing scientific software. On the other hand, the hybrid architectures (type B) for electrochemical modeling remains an open research question.** Type A is a practical near-term strategy for battery health forecasting, whereas type B will become dominant in the long term. The class of hybrid approaches have potential to fuse the causative and extrapolative capabilities of physics-based models with the speed, flexibility, and higher-dimensional capability of datadriven models, such as deep neural networks.*” In our opinion, the type B method is more essential, and it can truly integrate physical models and machine learning models. Our method belongs to category B2. By modeling the degradation dynamics model of the battery, the idea of PINN is introduced to solve the model. Moreover, we verified the feasibility and effectiveness of this method in transfer learning and small sample learning. To clarify how our proposed method differs from existing methods, and to naturally lead to our approach, we added the following statement to the revised manuscript:

”Although existing PINNs have shown promising results in battery modeling and prognosis, most of them focus on modeling within one cycle (such as state-of-charge (SOC) estimation and end-of-discharge (EOD) prediction). There are few publication that implement battery degradation modeling or use machine learning methods to solve the battery degradation equation to achieve SOH estimation. In this work, we proposed a PINN for battery SOH estimation. ...”

Figure 1: A screenshot of Fig. 1 from Aykol et al. [7].

Comment: Section 2 outlines their method implementation. It begins with the network architecture and data generation. In section 2.3, an essential aspect is discussed: the authors have chosen a 0.2 V range between the maximum charge voltage (or minimum discharge), but they don't clearly detail the conditions under which this measurement should occur (I miss specifications on the resting period required after full charge or discharge and its relation to the charging rate). This section needs more details and a better justification. In section 2.4, there's a brief comparison with a convolutional network. This could be enhanced by incorporating SOH estimation methods from other authors in the literature. Currently, they've shown that a neural architecture using physical knowledge outperforms a convolutional one, but this finding isn't very informative. When comparing network errors, it seems they are using features they've defined, essentially contrasting two algorithms they've developed, not benchmarking against the state-of-the-art in SOH data estimation. Moreover, they don't define the function \mathcal{F} incorporating the physical knowledge until section 4, making it necessary

to skip forward and then backtrack to understand these results. The document’s organization could be improved.

RESPONSE: Thanks for reviewer’s comment. We chosen a 0.2 V range between the maximum charge voltage for feature extraction. The default charging strategy is the charging process includes constant current and constant voltage (CC-CV) mode. The resting period after full charge or discharge is not necessary because our feature extraction process has nothing to do with the resting period. The charging rate has little impact on our feature extraction method, as long as the same battery is charged with the same strategy in all cycles. The four datasets we used all contain CC-CV charging mode, but the charging rate varies from dataset to dataset. The XJTU dataset and TJU dataset are charged in CC-CV mode throughout the entire process, while the HUST dataset and MIT dataset are first charged to 80% SOC and then charged in CC-CV mode. In this revised version, we have added statements to Section 2.3. *Feature extraction* in this revised manuscript.

“... We found that most datasets contain constant current and constant voltage (CC-CV) charging modes. No matter what strategy the battery is discharged with or whether it is fully discharged, it will eventually be fully charged in the CC-CV mode when charging.

Therefore, ... ”

The reviewer suggested that we compare the proposed method with other methods in the literature. Here, we make a discussion and clarification: In the field of computer vision (CV) or natural language processing (NLP), they have many benchmarks for comparison, and their comparison is fair (fixed database, unified data loading code, unified preprocessing code, similar model backbone). However, the field of battery health management lacks such benchmarks, and few authors make their code publicly available, which creates barriers to fair comparisons. While there are publicly available datasets for researchers to validate proposed methods, variations in the selection of training and test batteries, data preprocessing techniques, and the number of cycles after preprocessing introduce variability in model inputs. Consequently, achieving a fair comparison becomes challenging. This is also a pain point in the field of battery health management. In fact, we found that many methods proposed in some high-level articles are not compared with other methods [16; 17; 18; 19; 20] because it is impossible to create fair comparison conditions. Even though some articles compare their method with some other methods, these methods are also implemented by themselves [21; 22] based on their own data characteristics.

In addition, the main contribution of our manuscript is to provide a new perspective on battery modeling and prognosis that combines physical knowledge and machine learning, indicating the prospects of combining physical knowledge and machine learning for battery health management. What’s more, to ensure that the extracted features and the proposed PINN

have wider applicability, we extract features from the common parts contained in hundreds of charging and discharging protocols across 4 large-scale datasets. In contrast, many current studies only use a small amount of data to verify the proposed model. They extract specific features for specific datasets and are not very general. These methods may fail when changing other datasets.

In response to the reviewer’s comment, we still looked for test results of the other method proposed the other articles (**albeit with different model inputs and numbers of training and test batteries**). However, it seems that no method in any article has *been validated on all datasets we used*, especially our developed new dataset. Therefore, we can only look for different methods from different papers for each dataset, as shown in Table 1. For the above reasons, our method in our manuscript only compares MLP with the same backbone structure and CNNs with similar parameter amounts. They have the same data splitting method and preprocessing method, and the same input, ensuring relatively fair results.

Table 1: The Comparison between our proposed method and other methods.

Reference	Method	TJU						MIT		HUST	
		batch 1		batch 2		batch 3		—		—	
		MAPE	RMSE	MAPE	RMSE	MAPE	RMSE	MAPE	RMSE	MAPE	RMSE
Our paper	Our PINN	0.0164	0.0158	0.0119	0.0132	0.0080	0.0079	0.0065	0.0074	0.0078	0.0087
	MLP	0.0206	0.0197	0.0149	0.0157	0.0150	0.0144	0.0079	0.0087	0.0080	0.0090
	CNN	0.0198	0.0208	0.0143	0.0149	0.0124	0.0125	0.0065	0.0075	0.0074	0.0087
[21] ¹	ICA-based Linear model [23]	0.0130									
	RV-based Linear Model [24]	0.0250									
[25] ²	Proposed GPR			0.0216							
	LR [26]			0.0328							
	SVM [27]			0.0214							
	RFR [28]			0.0214							
	KNN [29]			0.0198							
[30]	LSTM [31]									0.0486	
	DCNN [32]									0.0413	
	Proposed method									0.0147	
[33]	MFL model							0.0104			
[34] ³	BTCN							0.0045			
[35]	SVR							0.0075	0.0102		
	MLP							0.0082	0.0109		
	RF							0.0072	0.0104		

Note: [1] The ICA refers to Incremental Capacity Analysis and the RV refers to Rest Voltage. "ICA-based Linear model [23]" means that this method was proposed in article [23] and cited and implemented in article [21] as a comparison method.

[2] In [25], only batteries cycled in the 25 degree and 45 degree are used.

[3] In [34], only 4 batteries were selected to validate the proposed method, and the results in the table are the average of 4 batteries.

Important: The training and test batteries used in these papers may be different from those in our manuscript, and the inputs to the models may also be different. Therefore, the above comparison is unfair and can only be used as a reference.

Finally, we apologize the issue of document’s organization with the manuscript that result in reviewer to skip forward and then backtrack to understand these results. This document’s organization is suggested by the journal template. A screenshot of the template is given in

Figure 2. If the modeling part is placed in an early Section, the Methods Section will appear abrupt and incomplete. Therefore, we cannot make major changes to the overall layout. In response to the reviewer’s suggestion and enable readers to see our description about the model earlier, we have made brief changes in Section 2.1. *Framework overview* and added a hyperlink to Section 4. Method in this Section:

”... The attributes that affect the battery degradation process are modeled from the perspective of empirical degradation and state space. Specifically, the proposed PINN consists of two parts: a solution function $u(\cdot)$ that maps the features to SOH and a dynamical function $\mathcal{F}(\cdot)$ that models battery degradation dynamic behaviors. We integrate the model with the neural network, and use the automatic differentiation mechanism of the neural network to solve our model (more details can be found in Section 4). ...”

ARTICLE SECTIONS

The text must be split into the sections given below. No other section headings are permitted.

Title

Authors

Abstract

Introduction

Results

Discussion (optional)

Methods (optional)

Data Availability

Code Availability (if applicable)

References

Acknowledgements (optional)

Author Contributions

Competing Interests

Figure 2: A screenshot of the template provided by Nature Communications.

Comment: In section 4, the authors introduce a function modeling physical knowledge about SOH’s temporal evolution. This section is somewhat challenging to follow. Equation (1) defines a SOH value sequence indexed by cycle. Equation 2 references a continuous variable $u(t)$; I gather u_k is a discretization of $u(t)$. The function \mathcal{F} is the derivative of $u(t)$ and depends on time, health, and a parameter. Equation 5 expands u to depend on a parameter x (though u is already defined as F ’s integral, thus depending on the previously introduced parameter Θ). Equation 6 redefines \mathcal{F} to depend on two parameters and also states u as a partial derivative of \mathcal{F} , deviating from equation (2). By equation (7), the explanation becomes muddled: it defines a function subtracting u_t (that was defined in equation (1) as a SOH sequence) from its partial derivative concerning time, indicating that u_t becomes a derivative. Yet, this implies

that, per definition (7), $f := 0$. Equation (8) redefines u_t to include u 's gradient concerning the parameter vector x ; by the same logic, I assume u_{xx} is this function's Hessian, though it doesn't seem to be used later. There are other issues of varying significance and other technical quality aspects, but at this point, the authors should revise this section and address these concerns before the manuscript can be considered.

RESPONSE: Thank you for your detailed observations and feedback on Section 4 of the manuscript. I appreciate your efforts in thoroughly examining the modeling section regarding the evolution of SOH over time. I am sorry that carelessness on our part has led to repeated use of symbols and lack of clarity in this section, causing you concerns about the clarity and consistency of the equations presented. Please allow me to address your points and provide clarification.

In the original version, we define $u_k = Q_k/Q_0$ as the SOH of a battery in the cycle k . Equation 2 references a continuous variable $u(t)$, and the reviewer is right that u_k is the discretization of $u(t)$. However, we use u_t in the following to represent the first partial derivative of u with respect to t , which causes it to be mixed with u_k . In this revised version, we use $u(k)$ to represent the SOH of the battery in cycle k to avoid duplication of its form with u_t , as show in Equation (1) (Equation (1) in Manuscript).

$$u(k) = \frac{Q_k}{Q_0}. \quad (1)$$

The discretized SOH sequence is defined as $u = [u(1), u(2), \dots, u(n)]$ (the original version is $u = [u_1, u_2, \dots, u_n]$), where n denotes the life cycles of a battery.

Equation (2) in the manuscript models the SOH degradation process as a time series, which is only the most basic form. However, this simple form is not sufficient to cover the dynamics during battery degradation. Therefore, Equation (2) needs to be extended (Equations (3) and (4) in the Manuscript) to take into account more factors. In this revised version, we added a introduction when presenting Equation (4) (Equation (2) in this Response Letter):

“ The Eq. 2 is reformulated as a form of partial differential equation (PDE) to incorporating more factors: ”

$$\frac{\partial u(t, \mathbf{x})}{\partial t} = \mathcal{F}(t, \mathbf{x}, u; \Theta). \quad (2)$$

The explanation of Equation (7) (Equation (5) in revised manuscript) becomes muddled because we define u_t as the partial derivative of u with respect to t ($u_t = \frac{\partial u(t, \mathbf{x})}{\partial t}$ in the Manuscript), which is formally the same as the SOH sequence defined in the original Equation (1). In this revised version, we redefine the SOH sequence, as shown in Equation (1).

For Equation 8 (Equation (6) in revised manuscript), we redefine u_t because the explicit form of battery degradation dynamics is not known, i.e., $\mathcal{F}(t, \mathbf{x}, u; \Theta)$ defined in Equation

(4) (Equation (2) in this Response Letter) is unknowable. Inspired by previous work [36; 37], since \mathcal{F} is not known anyway, we define a more general nonlinear dynamic form so that it can cover various situations [37], and use a neural network to model \mathcal{F} . Therefore, we get Equation (6) (Equation (3) in this Response Letter).

“... Inspired by [36; 37], we redefine a more general nonlinear dynamics \mathcal{F} of battery degradation, due to the fact that the explicit form of battery degradation is unknown:

$$u_t = \mathcal{F}(t, \mathbf{x}, u, u_t, u_{\mathbf{x}}, u_{\mathbf{x}\mathbf{x}}, \dots; \Theta), \quad (3)$$

The reviewer is right that $u_{\mathbf{x}\mathbf{x}}$ is a Hessian matrix. Considering the computational complexity, we discarded the high-order derivatives and only retained the influence of the first-order derivatives. Therefore, the final equation we modeled is:

$$u_t = \mathcal{F}(t, \mathbf{x}, u, u_t, u_{\mathbf{x}}; \Theta). \quad (4)$$

We also explain in our manuscript:

“... Finally, to balance the accuracy and computational complexity, we only consider the influence of the first-order partial derivative and discard the high-order derivative. ...”

I acknowledge that these equations require further clarification and refinement to accurately represent the battery degradation model. I am committed to revising this section comprehensively to enhance its clarity and coherence, ensuring a more precise description of the battery aging dynamics. Thank you for pointing out these important aspects that need improvement.

2 Response to Reviewer #2

There are three issues to address.

Thanks to reviewer for taking the time on our manuscript. We have read your comments in detail, thought deeply about them, and responded accordingly.

Comment: 1) The first, the plot in figure 4 did not properly convert to pdf. I can't see the plot. I tried different pdf viewers and all with the same results.

RESPONSE: Thanks for reviewer's comment. In our original manuscript, Figure 4 consists of sub-figures (a) and (b), and each sub-figure actually contains 16 sub-figures. We think the reason it cannot be converted to PDF correctly is that the dpi of the two figures is large. To ensure that the figure has a high enough resolution, we directly generate a PDF version with our code and save it. When we open sub-figure (a) or (b) alone on our computer, the computer takes a short time to load and freezes.

To answer the reviewer's question, we re-saved the two figures into "png" format and used them to generate this Response Letter. The Figure 4 in original manuscript (Figure 3a in revised manuscript). Since the Nature Communications limits the number of display items to 10, we have adjusted the figures and tables) is shown in Figure 3 (the revised manuscript still uses figures in PDF format).

Figure 3: An illustration of extracted features from the XJTU dataset batch 1. Different color represents different battery.

Comment: 2) I am not convinced that the good model performance is not simply due to information leakage. <https://www.sciencedirect.com/science/article/abs/pii/S2542435123003197>.

The current article doesn't ever really explain what the task of prediction is. Since SOH depends on both the battery internal quality as well as the us conditions, the linked article above argues that it is critical to be careful in defining the task you are trying to actually do (beyond the simple reply of "predict SOH"). What are you really trying to predict? Since the datasets were all created with different hypothesis in mind, it is hard to determine whether leakage is happening by using the same features across the data. The authors should more clearly define their task and clarify how they have ensured leakage is not the reason for good performance. Leakage can fool even careful practitioners.

RESPONSE: Thanks for reviewer's comment. Upon reviewing the link provided by the reviewer, we read in detail the article by Geslin et al. [38] titled *Selecting the appropriate features in battery lifetime predictions*, as well as the source article detailing the feature extraction methods employed in that study: Greenbank et al. [39] *Automated feature extraction and selection for data-driven models of rapid battery capacity fade and end of life*.

Let's start by answering the reviewer's first question: *The current article doesn't really explain what the task of prediction is?* The difference from the previous two articles is that our work focuses on state of health (SOH) estimation rather than remaining useful life (RUL) prediction. We give Figure 4 here to explain the difference between them. The SOH is defined as the ratio of the capacity of the current cycle to the initial capacity, and the RUL is typically defined as the remaining cycles that a lithium-ion battery is estimated to be functional [40]. Fundamentally, the SOH estimation task is a point-to-point regression problem, differing from the task addressed in Geslin's work.

Moreover, after a detailed review of Geslin's and Greenbank's articles, I found significant differences between our feature extraction method and theirs, if I understood correctly. In their studies, features derived from any one cycle are influenced by other cycles. Here, we present two screenshots from Greenbank's article, depicted as Figure 5 and Figure 6. They initially aggregate data across multiple cycles (Greenbank considers 9-19 cycles, Geslin mentions 10 cycles), as depicted in Figure 5(c). Subsequently, percentiles as shown in Figure 6 are derived, which are then utilized to compute features.

I suspect that the process of computing percentiles might lead to information leakage, as different charging strategies may result in disparate data distributions. Moreover, this process involves multiple cycles, potentially leading to the leakage of information from other cycles to the current one.

Additionally, the task of RUL prediction aims to answer the question: "Given historical observational data, how many cycles can the battery sustain under current charge-discharge

$$\text{SOH} = \frac{C_{max}(n)}{C_0} \times 100\%$$

$$\text{RUL} = n_{end} - n$$

Figure 4: The difference between SOH and RUL. C_0 represents initial (nominal) capacity, $C_{max}(n)$ represents the maximum available capacity of cycle n , and C_{fail} represents the capacity when the battery fails, which is generally defined as 80% or 70% of the nominal capacity [40].

protocols?” Hence, Geslin et al. argue that inconsistent charging protocols contribute to information leakage. Furthermore, they state in their article “**the CCCV at the end of charge is kept constant across all cells and mitigates explicit data leakage to the discharge data.**” This indicates that a fixed CCCV mode can reduce information leakage.

Returning to our study, our task involves SOH estimation on a cycle-to-point basis. In our feature extraction process, features from any given cycle are entirely independent of other cycles. Moreover, our feature extraction occurs during the CCCV stage before the battery reaches full charge, as shown in Figure 2b in the Manuscript File. Therefore, we believe our method should not encounter issues related to information leakage.

Finally, we genuinely appreciate the reviewer highlighting this potential concern. In future research, we will be more vigilant about this aspect.

Comment: 3) What the physics are that are being included is not really clear from the writing. It kind of exists in section 4, but it is hard to really figure out at a high level what physics are being accounted for. In an early section, a clearer description of the physical model(s) would be helpful.

RESPONSE: Thanks for reviewer’s comment. There are two types of tasks in the field of battery modeling and health management. One is a task within one charging/discharging cycle, including state-of-charge (SOC) estimation, end-of-discharge (EOD) prediction and other

Figure 5: A screenshot of Fig. 3 in Greenbank’s paper [39].

TABLE I
EXAMPLE FEATURE EXTRACTION VARIABLE BOUNDS FOR A TRAINING SET OF 50 CELLS

Percentile	Current [A]	Voltage [V]	Temperature [°C]	Power [W]
1 st	-4.00	2.00	30.0	-12.84
33 rd	-0.53	3.12	32.8	-1.08
67 th	1.00	3.51	35.3	3.43
99 th	6.00	3.60	40.3	21.34

Figure 6: A screenshot of Table 2 in Greenbank’s paper [39].

tasks. The other is the task in the entire life cycle, including SOH estimation, remaining-useful-life (RUL) prediction, degradation trajectory prediction. The dynamics of the battery in one cycle can be accurately described by physical models such as electrochemical models (pseudo-two-dimensional (P2D) model, single particle (SP) mode, etc.) and equivalent circuit models (ECM). However, it is very difficult to use electrochemical models to describe battery degradation over the life cycle because it involves many uncontrollable factors. Therefore, the current mainstream approach is to use empirical models to describe battery degradation process.

Some previous work proposed models, such as calendar and cycle aging model [10; 41], exponential degradation models [42; 43], failure forecast models (FFM) [44], etc., all of which

describe battery degradation from an empirical perspective. In fact, they also have difficulty explaining the physics at a high level. In this manuscript, we also model the attributes that affect the battery degradation process from the perspective of empirical degradation and state space, and use the idea of PINN to solve our model.

Regarding the reviewer’s suggestion that we describe the physical model in the early section, our manuscript’s layout is suggested by the journal template. A screenshot of the template is given in Figure 7. If the modeling part is placed in an early Section, the Methods Section will appear abrupt and incomplete. Therefore, we cannot make major changes to the overall layout. In response to the reviewer’s suggestion and enable readers to see our description about the model earlier, we have made brief changes in Section *2.1. Framework overview* and added a hyperlink to Section *4. Method* in this Section.

“... The attributes that affect the battery degradation process are modeled from the perspective of empirical degradation and state space. Specifically, the proposed PINN consists of two parts: a solution function $u(\cdot)$ that maps the features to SOH and a dynamical function $\mathcal{F}(\cdot)$ that models battery degradation dynamic behaviors. We integrate the model with the neural network, and use the automatic differentiation mechanism of the neural network to solve our model (more details can be found in Section 4). ...”

ARTICLE SECTIONS

The text must be split into the sections given below. No other section headings are permitted.

Title

Authors

Abstract

Introduction

Results

Discussion (optional)

Methods (optional)

Data Availability

Code Availability (if applicable)

References

Acknowledgements (optional)

Author Contributions

Competing Interests

Figure 7: A screenshot of the template provided by Nature Communications.

3 Response to Reviewer #3

Comment: In this manuscript, neural networks are used to estimate the SOH from statistical parameters during a period of (full) charge. This first neural network is then used as the input of a second PINN to estimate the progression of the SOH over time. The application to different, comprehensive datasets of cells with different chemistry, make the work interesting, especially the application of transfer learning. We suggest accepting the manuscript after major revisions, as some work needs to be done to make the manuscript more readable and to provide additional information on parameterization and selection of hyperparameters to make this a valuable contribution to the community. We would recommend adapting the structure of the manuscript so that the method is explained first, then the results are presented and discussed and finally there should be a conclusion section. Special attention should be paid to avoid jumping between the manuscript and supplementary material. Thus, all relevant information, especially on the features, should also be included in the manuscript. Please consider the following remarks:

RESPONSE: Thank you for your thoughtful review and constructive feedback on our manuscript.

We appreciate your positive comments on the significance of our work and your valuable suggestions on the need to further improve the readability of the manuscript and provide additional information on parameterization and hyperparameter selection. Regarding the structure of the manuscript, we understand the importance of presenting our methods, results, and conclusions clearly and organically. We will reorganize the manuscript to first provide a comprehensive explanation of our methods, followed by presentation and discussion of the results, ensuring that information is coherent and not spread between the main text and the supplementary material. Specifically, we will work to incorporate these suggestions into the revised manuscript and provide a point-by-point response below, outlining the changes made in response to your comments.

Comment: 1. In the abstract, please name the error metric (MAE, RMSE) you refer your final validation results to.

RESPONSE: Thanks to reviewer’s comment. The error in Abstract refers to MAPE (Mean Absolute Percentage Error), which is calculated based on the 4 values (0.85%, 1.21%, 0.65%, and 0.78%) in Section *2.4 SOH estimation*. In this revised manuscript, we point out the error metric in the Abstract.

“... The mean absolute percentage error (MAPE) is 0.87%. ...”

Comment: 2. In the Introduction, please give more details about your chosen PINN ar-

chitecture with reference to other works, i.e. , *Aykol et al. (10.1149/1945-7111/abec55)* or *Hofmann et al. (10.1149/1945-7111/acf0ef)*. With the current introduction, it does not get directly clear where the physics-based modelling takes place. It more looks like it’s a sequential NN with a novel feature extraction method.

RESPONSE: Thanks for reviewer’s comment. When we were conducting the literature review, we had read in detail the article by Aykol et al., as cited in the reference [36] in the **original manuscript**. The article by Aykol et al. gives five approaches that combine physical knowledge and machine learning for battery modeling, as shown in Figure 1. They are: A1. Sequential Integration: Residual or Delta Learning; A2. Sequential Integration: Transfer Learning; A3. Sequential Integration: Parameter Learning; B1.Hybrid: Physics-constrained MLM; B2.Hybrid: ML-accelerated PBM.

In the article by Hofmann et al., they used the P2D model to generate simulation data and combined it with laboratory data and vehicle field data to train the neural network model. The model they finally obtained is shown in Figure 8. According to our understanding, the model they proposed belongs to a type of Sequential PINN (A1 or A2). The physical knowledge is reflected in the use of immeasurable internal states, such as concentrations and potentials.

Figure 8: A screenshot of Fig. 8 in Hofmann’s article [12].

Aykol et al. said in the paper that the architecture of type B2 relies on physics-informed ML. This hidden physics models have been pioneered by Raissi and co-workers, leading to the design of physics-informed neural networks (PINNs) [36]. PINNs exploit the automatic

differentiation and universal function approximator aspects of neural networks to train on, solve and/or discover the nonlinear PDEs of the observed system, with small amounts of data, in effect obeying the underlying physical laws and boundary conditions.

According to the above description, our model should belong to category B2. One fact is that the process of a battery during a charge and discharge cycle can be described by the electrochemical PDE equation (such as the P2D model); however, during the entire life cycle, as far as we know, there does not seem to be an equation that can accurately describe the battery degradation. Some previous work proposed models, such as calendar and cycle aging model [10; 41], exponential degradation models [42; 43], failure forecast models (FFM) [44], etc., all of which describe battery degradation from an empirical perspective. In this manuscript, we also refer to the idea of PINN from the perspective of empirical degradation and state space, model the attributes that affect the battery degradation process. We utilized the automatic differentiation and universal function approximators of deep learning to design a simple neural network structure to solve the degradation model we established. In the process of designing and solving the our model, we have referred to the ideas of Raissi et al. [36; 37], as described in Section 4.2 *Physics-informed neural network* of the Manuscript File. In this revised manuscript, we adopted the reviewer’s suggestion and added an description to the proposed PINN in the Introduction.

“... Aykol et al. [7] presented 5 approaches that combine physical knowledge and machine learning for battery modeling. ...”

“ Although existing PINNs have shown promising results in battery modeling and prognosis, most of them focus on modeling within one cycle (such as state-of-charge estimation and end-of-discharge prediction). There are few publication that implement battery degradation modeling or use machine learning methods to solve the battery degradation equation to achieve SOH estimation. In this work, we proposed a PINN for battery SOH estimation. ...”

Comment: 3. Please include the paper by *Hofmann et al.* (10.1149/1945-7111/acf0ef) into your reference list as they have also introduced a sequential PINN for SOH estimation.

RESPONSE: Thanks for reviewer’s comment. In this revised version, we have cited the article by Hofmann et al. (10.1149/1945-7111/acf0ef).

“... Hofmann et al. [12] used the pseudo-two-dimensional (P2D) Newman model to generate data at different health status points and combined it with experimental data and field data to train the neural network model, which takes advantage of the correlation between internal states and measurable SOH. ”

Comment: 4. In Figure 2 please provide the units in your axis-labels, i.e., “Capacity /

Ah". Please include a legend to allow mapping every batch to the specific degradation curves. Further, make the Figure bw-readable by using different linestyles.

RESPONSE: Thanks to the reviewer for pointing out this issue. In this revised manuscript, we have added units and legends, and marked different batches with different markers, as shown in Figure 9 (Figure 2a in revised manuscript).

Figure 9: The degradation trajectories of the XJTU battery dataset. We use different colors to represent different batches.

Comment: 5. We understand that the statistical features are all connected to the end of the charge process and this approach is possible for all applications, where a full charge is performed frequently. However, it is not true that for all or even most applications a full charge is performed frequently. In contrary users are directed to stop charging at 80% by special apps to avoid increased aging.

RESPONSE: Thanks for reviewer’s comment. As far as we know, many devices we are familiar with are often fully charged during their charging stages, such as electric vehicles [45; 46], unmanned aerial vehicles [47], mobile phones, and satellites [48]. Full charging provides extended range for electric vehicles and unmanned aerial vehicles, and longer use time for phones. In the case of satellites, batteries are often operated at shallow discharge depths for safety and reliability reasons, and remain fully charged most of time [49]. Additionally, many published works [21; 50; 45; 22; 48] have proposed their methods based on batteries being fully charged.

However, we agree that the reviewer’s concern is valid. Some device designers and manufacturers may offer users the option to set battery levels at 80%, allowing users to customize based on preference. Our method indeed does not cover this particular scenario. For this scenario, new feature extraction rules need to be designed, but the PINN model framework we proposed can still be used. We greatly appreciate the reviewer highlighting this point, and we will pay more attention to such situations in our future research.

Comment: 6. It is claimed, that “An important finding is that the magnitude of the correlation coefficient between each feature and SOH is related to the chemical composition of a battery and is less affected by the charge/discharge protocols.” However, there is no evidence given. Please refrain from that claim or provide evidence.

RESPONSE: Thank you for your comment. We acknowledge that the reviewer is very rigorous and I apologize for our lack of rigor. The statement in the manuscript is a description or conjecture given by us based on our experimental observations. It cannot be called a “finding” due to the lack of sufficient evidence. In this revised manuscript, to be more rigorous, we changed it to a conjecture:

“Based on experimental phenomena and analysis of Figure 3b, we give a natural conjecture: the magnitude of the correlation coefficient between each feature and SOH is related to the chemical composition of a battery and is less affected by the charge/discharge protocols. ...”

Comment: 7. Please extend Figure 3 to also show which time-series data is additionally used for SOH estimation. From the text and Figures it gets very clear which data is used for feature extraction, but it is hard to grasp which time-series data is additionally used for SOH estimation together with the extracted features.

RESPONSE: Thank you for your comment. In Figure 3 of original (Figure 2b in revised manuscript. Since the Nature Communications limits the number of display items to 10, we have adjusted the figures and tables), we have shown the data used to feature extraction. A total of 16 features were extracted from voltage and current curves. When modeling, the extracted 16 features and cycle number were used to estimate SOH (a 17-dimensional vector). Since the cycle is not easily shown clearly in the figure, we explained it in Section 2.4 *SOH estimation*.

2.4. *SOH estimation*

The extracted 16 features and time (cycle) are used as inputs of the proposed PINN to estimate SOH. ...

Comment: 8. In Figure 5 please make clear which feature number refers to which feature. If the order is consistent with Figure 4, please mention that again. In the text, also include that you have used the Pearson correlation coefficient.

RESPONSE: Thanks to the reviewer for pointing out this issue. The order of features in Figure 5 (Figure 3b in revised manuscript) is consistent with that in Figure 4 (Figure 3a in revised manuscript), please forgive us for forgetting to mention it again. In this revised manuscript, we mention it in the caption of Figure 3b (Figure 10 in this Response Letter). At

the same time, we also indicate that we have used the Pearson correlation coefficient in the article.

2.3. Feature extraction

”... Further, we extracted features from 387 batteries in 4 datasets respectively, and calculated the Pearson correlation coefficient between features and SOH within each dataset, ...”

Figure 10: An illustration of extracted features and correlation coefficients. (a). Features from the XJTU dataset batch 1. Different colors represent different batteries. (b). Correlation heatmap between extracted features and SOH in 4 datasets. The numbers 1 to 16 represent 16 features, and the order of features is consistent with that in (a).

Comment: 9. In figure 7 the y-axis should be scaled equal for all subplots for better comparability of the performance with respect to different datasets. Again, errors given in % are more readable than fractions (i.e. 1% instead 0.01).

RESPONSE: Thanks for reviewer’s comment. In this revised manuscript, we set the y-axis of each subplot in % format to make it more readable. However, we think that different datasets have varying estimation complexities. It is unnecessary to forcibly set the y-axis of each subplot to be equal, as it can be unfriendly to the presentation. In addition, by setting the y-axis to the % format, it is easier to see the y-axis range of each subplot, and it is relatively easy to compare the performance with respect to different datasets. Therefore, we still retained the original y-axis in the revised manuscript. In this Response Letter, we presented two situations to support our above statement, as shown in Figure 11.

However, we do not think this is a controversial issue. If the reviewer still insists that the y-axis should be scaled equal, we can also put Figure 11b into the manuscript.

(a)

(b)

Figure 11: An illustration of test error distributions for 3 models on 4 datasets. (a): the raw version. (b): the version that all y-axes are set equal.

Comment: 10. Please comment on the different correlations of the statistical features with respect to chemistry. It is not obvious why there should be a difference between NMC and LFP cells.

RESPONSE: Thanks for reviewer's comment. For this problem, we conducted some analysis from the perspective of data, as shown in Figure 12. In the figure, we draw the current and voltage curve segments used for feature extraction in 4 datasets. Among them, the electrochemical composition of the XJTU batteries is $\text{LiNi}_{0.5}\text{Co}_{0.2}\text{Mn}_{0.3}\text{O}_2$, the TJU batteries is $\text{Li}_{0.84}\text{Ni}_{0.83}\text{Co}_{0.11}\text{Mn}_{0.07}\text{O}_2$. We will call them NCM battery in the following. The the electrochemical composition of MIT batteries and HUST batteries are both LiFePO_4 (LFP).

Some interesting phenomena can be observed from the figure. For NCM batteries, as the battery ages, the constant voltage (CV) charging time gradually increases (that is, the time it takes for the current to drop from 0.5 A to 0.1 A becomes longer) (Figure 12(b) and Figure 12(d)). However, for LFP batteries, the phenomenon is opposite: as the battery ages, the CV charging time gradually becomes shorter (Figure 12(f) and Figure 12(h)). Among the extracted features, the time it takes for the current to drop from 0.5 A to 0.1 A is one of the features ("CV charge time" in Figure 3a of the Manuscript (Figure 10a in this Response Letter), feature numbered 14 in Figure 3b of the Manuscript (Figure 10b in this Response Letter)). For NCM batteries, the charging time becomes longer, so "CV charge time" shows a negative correlation with SOH, and the correlation is strong. For LFP batteries, the charging time shorter, and "CV charge time" shows a positive correlation with SOH. They are consistent with the results in Figure 10b. Other features derived from current curves can presumably be interpreted similarly.

For the voltage curves in the constant current (CC) charging stage (the voltage range is $[V_{\text{end}-0.2}, V_{\text{end}}]$), the charging time of LFP batteries (Figure 12(e) and Figure 12(g)) obviously gradually becomes shorter as the battery ages, so the "CC charge time" shows a high positive correlation with SOH (feature numbered 6 in Figure 10b). The voltage change trend of the NCM batteries does not seem to be as obvious as that of the LFP batteries from the Figure 12(a) and Figure 12(c). Therefore, LFP batteries are superior to NCM batteries in terms of consistency of all features derived from voltage curves.

The above are some phenomena we observed and some conjectures we got from the changes in the voltage and current curve. There should be deeper reasons behind these phenomena, such as the chemical composition of the battery and other factors. However, exploring the above reasons from the perspective of battery chemical composition is beyond our ability in this paper. The deeper reasons may require further research by more scholars. For the sake of rigor, and based on the reviewer's **Comment** 6, we have revised the sentence in this revised manuscript that describes the correlation coefficient between the features and SOH with the

connection to battery chemical components.

“Based on experimental phenomena and analysis of Figure 5, we give a natural conjecture: the magnitude of the correlation coefficient between each feature and SOH is related to the chemical composition of a battery and is less affected by the charge/discharge protocols. ...”

Figure 12: Schematic diagram of current and voltage curves used to extract features in 4 datasets. Among them, XJTU battery dataset and TJU dataset come from NCM batteries, and MIT dataset and HUST dataset come from LFP batteries.

Comment: 11. As some features seem to be highly correlated, could you please comment on the selection of the features? How did you decide to incorporate all 16 features?

RESPONSE: Thanks for reviewer’s comment. The reviewer raises a pertinent point regarding the high correlation among some features. However, our decision to incorporate all 16 features is informed by several considerations.

Primarily, previous studies [21; 51; 52; 53] in the field of battery health management often employ multiple features. Given the adaptability of neural network models to handle higher-dimensional inputs, the inclusion of 16 features is within a reasonable range. Hence, we refrained from conducting feature selection.

Moreover, aiming for a more generalized SOH estimation model led us to retain all features. While certain features might demonstrate high correlation, each feature potentially encapsulates unique aspects of SOH variation. Considering the potential diversity across datasets, a smaller feature set might excel in one dataset but falter in others, hence compromising the model’s generalizability.

Additionally, noise in the dataset is a crucial concern. A reduced feature set might exacerbate susceptibility to noise interference. By incorporating all 16 features, we aimed to fortify the model against potential noise impacts without sacrificing predictive performance significantly.

Furthermore, the possibility of interaction effects among features is crucial. Even correlated features might contribute to the prediction through their combined or interactive effects, and eliminating them might undermine the model’s capability to capture these complex relationships.

In summary, our decision to include all 16 features stems from precedents set by similar studies, the adaptability of neural networks to higher dimensions, the pursuit of a universally applicable model across diverse datasets, the need for robustness against noise, and the consideration of potential interaction effects among features.

Comment: 12. In Section 2.4 please include more information about the data split. How did you make sure that batteries from the same batch are not too similar and hence the test datasets is too similar to the training dataset? How much percentage is training and how much is test data?

RESPONSE: Thanks for reviewer’s feedback. In the initial version of the Supplementary file, we provided the number of batteries used for testing in each dataset in Table S.2. Within each dataset, the ratio of training, validation, and testing sets is approximately divided in a ratio close to 6:2:2, although it is not strictly adhered to. For instance, the XJTU batch 1 dataset comprising 8 batteries, we selected 2 batteries as the test set (the test battery ratio

is 25%). From the remaining 6 batteries, we first shuffled all cycle data for each battery, then allocated 20% of the data as validation sets, and the rest as training sets. To ensure equitable data allocation, we selected data from batteries numbered 4 and 8; this methodology was maintained across other batches (as shown in Table S.3). Similarly, the MIT dataset comprising 125 batteries, we selected batteries with IDs as multiples of 5 for the test set (23 batteries), aiming to ensure a more even distribution of test batteries across the entire dataset. In response to the reviewer’s suggestion, we have included additional clarification in Section 2.4. *SOH estimation* in this revised manuscript.

“... For each dataset, we divide the training battery, validation battery, and test battery approximately in a ratio of 6:2:2. The number of test batteries in each dataset can be found in Table S2. To ensure fairness, the numbers of test batteries are evenly distributed throughout the dataset, as shown in Tables S.3 to S.6. ...”

Comment: 13. Please provide information about the hyperparameter tuning for the proposed PINN. Which architectures were explored and which approach (e.g. Bayes, Random Search, Grid Search) was used for optimizing the hyperparameters.

RESPONSE: Thanks to reviewer’s comment. In the process of hyperparameter tuning, we used Grid Search method to optimize the hyperparameters, including the number of PINN layers, the number of neurons in each layer, and the trade-off parameters α and β . Other parameters, such as batch size, have relatively little impact on model performance, so they are designed based on experience, computer performance, data volume, and other information. For the learnable parameters in PINN, the Adam optimizer is used to optimize. In this revised version, we have added relevant information in Supplementary Note 3.

“... The proposed PINN was learned by minimizing the loss defined in Eq. 13 of the Manuscript File, and the Adam optimizer was used in the training phase. In the process of hyperparameter tuning, the Grid Search strategy was adopted to optimize the hyperparameter, including the number of PINN layers, the number of neurons in each layer, and α and β . The trade-off parameters α and β are set to 0.7 and 20 for XJTU battery dataset, 1 and 50 for TJU and MIT datasets, and 0.5 and 80 for HUST dataset. ...”

Comment: 14. Please provide an updated benchmark to create a fair comparison of the PINN to the MLP and CNN. This means, that the hyperparameters of the MLP and CNN should be tuned to their respective optima. In this matter, also include the computational complexity of all three models.

RESPONSE: Thanks for reviewer’s comment. In this revised manuscript, we have optimized the parameters of MLP and CNN again so that they can be adjusted to optimal values. We

use the Adam optimizer and use the cosine annealing algorithm to control the learning rate. The optimized results were used to update Table 2 in the Manuscript (Table 2 in this Response Letter) and Tables S.3, S.4, S.5, and S.6 in the Supplementary material.

Table 2: The results of proposed PINN (Ours), MLP, and CNN on 4 datasets. MAPE is the mean absolute percentage error, and RMSE is the root mean square error. The best results are shown in bold. All values are averaged from 10 experiments.

Dataset	Batch	Ours		MLP		CNN	
		MAPE	RMSE	MAPE	RMSE	MAPE	RMSE
XJTU	1	0.0070	0.0094	0.0260	0.0277	0.0270	0.0330
	2	0.0113	0.0122	0.0275	0.0304	0.0298	0.0352
	3	0.0086	0.0100	0.0211	0.0237	0.0177	0.0212
	4	0.0071	0.0105	0.0200	0.0235	0.0150	0.0189
	5	0.0105	0.0135	0.0183	0.0217	0.0350	0.0453
	6	0.0063	0.0097	0.0204	0.0242	0.0149	0.0194
TJU	1	0.0164	0.0158	0.0206	0.0197	0.0198	0.0208
	2	0.0119	0.0132	0.0149	0.0157	0.0143	0.0149
	3	0.0080	0.0079	0.0150	0.0144	0.0124	0.0125
MIT	—	0.0065	0.0074	0.0079	0.0087	0.0065	0.0075
HUST	—	0.0078	0.0087	0.0080	0.0090	0.0074	0.0087

In addition, we also calculated the inference time of all three models, as shown in Table S.8 in the Supplementary material (Table 3 in this Response Letter). Since the number of parameters of all three models is small, we did not use GPU for acceleration. Three models were implemented in Pytorch 1.7.1 on Intel Core i5-10400F CPU @ 2.90 GHz. It can be seen from the Table 3, the inference speed of all three models is very fast. The solution $u(\cdot)$ of our PINN is the same as MLP. During the forward inference process, dynamics $\mathcal{F}(\cdot)$ does not need to be calculated, so the inference time of our PINN is similar to MLP.

Table 3: The details of proposed PINN, MLP, and CNN. Sin refers to the sine function. BasicBlock is similar to that in ResNet [54], which consists of Conv1d, BatchNorm1d, ReLU, Conv1d, and BatchNorm1d.

Model	Module	Layer	Input size	Output size	Inference Param. Num.	Inference time/1000 sample
PINN	$u(\cdot)$	Linear+Sin	17	60	7781	5.81e-04
		Linear+Sin	60	60		
		Linear	60	32		
		Linear+Sin	32	32		
		Linear	32	1		
	$\mathcal{F}(\cdot)$	Linear+Sin	35	60		
		Linear+Sin	60	60		
		Linear	60	1		
	MLP		Linear+Sin	17	60	7781
Linear+Sin			60	60		
Linear			60	32		
Linear+Sin			32	32		
Linear			32	1		
CNN		BasicBlock	(1,17)	(8,17)	8465	1.29e-02
		BasicBlock	(8,17)	(16,9)		
		BasicBlock	(16,9)	(24,5)		
		BasicBlock	(24,5)	(16,5)		
		BasicBlock	(16,5)	(8,5)		
		Linear	8*5	1		

Note: The values in the "Inference time/1000 sample" column represents the the time, in seconds, spent in inference per 1000 samples. Specifically, we set the batch size to 1000, count the time spent on 1000 forward inferences, and then take the average. Since the number of parameters of three model is small, we do not use GPU for acceleration. Three models were implemented in Pytorch 1.7.1 on Intel Core i5-10400F CPU @ 2.90 GHz.

Comment: 15. Please expand the explanations on the PINN: which aspects in this model are to be understood as physical boundary conditions and which are not? In particular, since the partial derivatives of the SOH with respect to the statistical features are included, it is not obvious which physical, chemical boundary conditions are to be considered here.

RESPONSE: Thanks to reviewer’s comment. PINN was originally proposed in the paper [36] and is used to solve partial differential equations (PDE), such as Schrödinger equation and Navier-Stokes equation. In these equations, they have obvious boundary conditions in mathematical formulas and have physical meanings. However, in the field of battery SOH estimation, as the reviewer said, the boundary conditions are not obvious. We model the battery from the perspective of degradation dynamics, which essentially does not involve the physical and chemical changes inside the battery. Therefore, we do not consider chemical boundary conditions, but only the initial boundary condition and the monotonicity of the battery degradation process.

Ideally, $u(0) = 1$. In fact, $u(0)$ is not equal to 1 in many cases, and it can be calculated as $u(0) = Q_{\text{initial}}/Q_0$, where Q_{initial} represents initial capacity, and Q_0 denotes nominal capacity.

In addition, the battery degradation process should be monotonic, that is, the SOH of the next cycle should be less than or equal to the previous cycle, unless capacity regeneration occurs. Therefore, we design a monotonicity loss to constrain the model during the training stage:

$$\mathcal{L} = \mathcal{L}_{\text{data}} + \alpha\mathcal{L}_{\text{PDE}} + \beta\mathcal{L}_{\text{mono}}, \quad (5)$$

If capacity regeneration occurs, the trade-off parameter β can be adjusted to change the impact of $\mathcal{L}_{\text{mono}}$ on model performance.

4 Response to Reviewer #4

Comment: I co-reviewed this manuscript with one of the reviewers who provided the listed reports. This is part of the Nature Communications initiative to facilitate training in peer review and to provide appropriate recognition for Early Career Researchers who co-review manuscripts.

RESPONSE: Thank you for your valuable contribution as a co-reviewer in evaluating our manuscript. We highly value the insights and perspectives shared by both you and the other reviewers. Your comments have significantly contributed to the enhancement of our manuscript. Once again, we extend our gratitude for your contribution and constructive feedback, which will undoubtedly help us in refining our manuscript to meet the required standards for publication.

Finally, we also found some annotations on our manuscript, which we suspect may have been annotated by you or other reviewers. We have modified every annotation. Thanks again!

References

- [1] H. Rauf, M. Khalid, N. Arshad, Machine learning in state of health and remaining useful life estimation: Theoretical and technological development in battery degradation modelling, *Renewable and Sustainable Energy Reviews* 156 (2022) 111903.
- [2] Y. Che, X. Hu, X. Lin, J. Guo, R. Teodorescu, Health prognostics for lithium-ion batteries: mechanisms, methods, and prospects, *Energy & Environmental Science* (2023).
- [3] T. F. Fuller, M. Doyle, J. Newman, Simulation and optimization of the dual lithium ion insertion cell, *Journal of the electrochemical society* 141 (1) (1994) 1.
- [4] X. Liu, X.-Q. Zhang, X. Chen, G.-L. Zhu, C. Yan, J.-Q. Huang, H.-J. Peng, A generalizable, data-driven online approach to forecast capacity degradation trajectory of lithium batteries, *Journal of Energy Chemistry* 68 (2022) 548–555.
- [5] Z. Chen, Y. Liu, H. Sun, Physics-informed learning of governing equations from scarce data, *Nature communications* 12 (1) (2021) 6136.
- [6] G. E. Karniadakis, I. G. Kevrekidis, L. Lu, P. Perdikaris, S. Wang, L. Yang, Physics-informed machine learning, *Nature Reviews Physics* 3 (6) (2021) 422–440.
- [7] M. Aykol, C. B. Gopal, A. Anapolsky, P. K. Herring, B. van Vlijmen, M. D. Berliner, M. Z. Bazant, R. D. Braatz, W. C. Chueh, B. D. Storey, Perspective—combining physics and machine learning to predict battery lifetime, *Journal of The Electrochemical Society* 168 (3) (2021) 030525.
- [8] R. G. Nascimento, F. A. Viana, M. Corbetta, C. S. Kulkarni, A framework for li-ion battery prognosis based on hybrid bayesian physics-informed neural networks, *Scientific Reports* 13 (1) (2023) 13856.
- [9] A. Thelen, Y. H. Lui, S. Shen, S. Laflamme, S. Hu, H. Ye, C. Hu, Integrating physics-based modeling and machine learning for degradation diagnostics of lithium-ion batteries, *Energy Storage Materials* 50 (2022) 668–695.
- [10] J. Shi, A. Rivera, D. Wu, Battery health management using physics-informed machine learning: Online degradation modeling and remaining useful life prediction, *Mechanical Systems and Signal Processing* 179 (2022) 109347.

- [11] F. Wang, Q. Zhi, Z. Zhao, Z. Zhai, Y. Liu, H. Xi, S. Wang, X. Chen, Inherently interpretable physics-informed neural network for battery modeling and prognosis, *IEEE Transactions on Neural Networks and Learning Systems* (2023). doi:<https://doi.org/10.1109/TNNLS.2023.3329368>.
- [12] T. Hofmann, J. Hamar, M. Rogge, C. Zoerr, S. Erhard, J. P. Schmidt, Physics-informed neural networks for state of health estimation in lithium-ion batteries, *Journal of The Electrochemical Society* 170 (9) (2023) 090524.
- [13] J. Tian, R. Xiong, J. Lu, C. Chen, W. Shen, Battery state-of-charge estimation amid dynamic usage with physics-informed deep learning, *Energy Storage Materials* 50 (2022) 718–729.
- [14] H. Tu, S. Moura, Y. Wang, H. Fang, Integrating physics-based modeling with machine learning for lithium-ion batteries, *Applied Energy* 329 (2023) 120289.
- [15] X. Jia, C. Zhang, Y. Li, C. Zou, X. Cai, et al., Knee-point-conscious battery aging trajectory prediction based on physics-guided machine learning, *IEEE Transactions on Transportation Electrification* (2023).
- [16] W. Li, J. Chen, K. Quade, D. Luder, J. Gong, D. U. Sauer, Battery degradation diagnosis with field data, impedance-based modeling and artificial intelligence, *Energy Storage Materials* 53 (2022) 391–403.
- [17] J. Tian, R. Xiong, W. Shen, J. Lu, X.-G. Yang, Deep neural network battery charging curve prediction using 30 points collected in 10 min, *Joule* 5 (6) (2021) 1521–1534.
- [18] K. A. Severson, P. M. Attia, N. Jin, N. Perkins, B. Jiang, Z. Yang, M. H. Chen, M. Aykol, P. K. Herring, D. Fraggedakis, et al., Data-driven prediction of battery cycle life before capacity degradation, *Nature Energy* 4 (5) (2019) 383–391.
- [19] Y. Zhang, Q. Tang, Y. Zhang, J. Wang, U. Stimming, A. A. Lee, Identifying degradation patterns of lithium ion batteries from impedance spectroscopy using machine learning, *Nature communications* 11 (1) (2020) 1706.
- [20] P. K. Jones, U. Stimming, A. A. Lee, Impedance-based forecasting of lithium-ion battery performance amid uneven usage, *Nature Communications* 13 (1) (2022) 4806.

- [21] J. Zhu, Y. Wang, Y. Huang, R. Bhushan Gopaluni, Y. Cao, M. Heere, M. J. Mühlbauer, L. Mereacre, H. Dai, X. Liu, et al., Data-driven capacity estimation of commercial lithium-ion batteries from voltage relaxation, *Nature communications* 13 (1) (2022) 2261.
- [22] J. Lu, R. Xiong, J. Tian, C. Wang, F. Sun, Deep learning to estimate lithium-ion battery state of health without additional degradation experiments, *Nature Communications* 14 (1) (2023) 2760.
- [23] P. Pei, Q. Zhou, L. Wu, Z. Wu, J. Hua, H. Fan, Capacity estimation for lithium-ion battery using experimental feature interval approach, *Energy* 203 (2020) 117778.
- [24] I. Baghdadi, O. Briat, P. Gyan, J. M. Vinassa, State of health assessment for lithium batteries based on voltage–time relaxation measure, *Electrochimica Acta* 194 (2016) 461–472.
- [25] C.-J. Ko, K.-C. Chen, T.-W. Su, Differential current in constant-voltage charging mode: A novel tool for state-of-health and state-of-charge estimation of lithium-ion batteries, *Energy* (2023) 129826.
- [26] X. Sui, S. He, S. B. Vilsen, J. Meng, R. Teodorescu, D.-I. Stroe, A review of non-probabilistic machine learning-based state of health estimation techniques for lithium-ion battery, *Applied Energy* 300 (2021) 117346.
- [27] X. Sui, S. He, J. Meng, R. Teodorescu, D.-I. Stroe, Fuzzy entropy-based state of health estimation for li-ion batteries, *IEEE Journal of Emerging and Selected Topics in Power Electronics* 9 (4) (2020) 5125–5137.
- [28] K. S. Mawonou, A. Eddahech, D. Dumur, D. Beauvois, E. Godoy, State-of-health estimators coupled to a random forest approach for lithium-ion battery aging factor ranking, *Journal of Power Sources* 484 (2021) 229154.
- [29] Y. Zhou, M. Huang, M. Pecht, Remaining useful life estimation of lithium-ion cells based on k-nearest neighbor regression with differential evolution optimization, *Journal of Cleaner Production* 249 (2020) 119409.
- [30] Y. Ge, J. Ma, G. Sun, Health status prediction of lithium ion batteries based on zero-shot learning, *Journal of Energy Storage* 72 (2023) 108494.
- [31] K. Park, Y. Choi, W. J. Choi, H.-Y. Ryu, H. Kim, Lstm-based battery remaining useful life prediction with multi-channel charging profiles, *Ieee Access* 8 (2020) 20786–20798.

- [32] S. Shen, M. Sadoughi, M. Li, Z. Wang, C. Hu, Deep convolutional neural networks with ensemble learning and transfer learning for capacity estimation of lithium-ion batteries, *Applied Energy* 260 (2020) 114296.
- [33] C. Zhang, S. Li, Y. Shang, B. Duan, K. Liu, Z. Bi, G. Zhao, C. Li, N. Wang, State of health estimation for lithium-ion batteries based on mechanism fundamental learning under variable charging strategies (2023).
- [34] J. Gao, D. Yang, S. Wang, Z. Li, L. Wang, K. Wang, State of health estimation of lithium-ion batteries based on mixers-bidirectional temporal convolutional neural network, *Journal of Energy Storage* 73 (2023) 109248.
- [35] H. Hamed, M. Yusuf, M. Suliga, B. Ghalami Choobar, R. Kostos, M. Safari, An incremental capacity analysis-based state-of-health estimation model for lithium-ion batteries in high-power applications, *Batteries & Supercaps* (2023) e202300140.
- [36] M. Raissi, P. Perdikaris, G. E. Karniadakis, Physics-informed neural networks: A deep learning framework for solving forward and inverse problems involving nonlinear partial differential equations, *Journal of Computational physics* 378 (2019) 686–707.
- [37] M. Raissi, Deep hidden physics models: Deep learning of nonlinear partial differential equations, *The Journal of Machine Learning Research* 19 (1) (2018) 932–955.
- [38] A. Geslin, B. van Vlijmen, X. Cui, A. Bhargava, P. A. Asinger, R. D. Braatz, W. C. Chueh, Selecting the appropriate features in battery lifetime predictions, *Joule* 7 (9) (2023) 1956–1965.
- [39] S. Greenbank, D. Howey, [Automated feature extraction and selection for data-driven models of rapid battery capacity fade and end of life](https://ieeexplore.ieee.org/document/9520291), *IEEE Transactions on Industrial Informatics* 18 (5) (2021) 2965–2973.
URL <https://ieeexplore.ieee.org/document/9520291>
- [40] Y. Zhang, Y.-F. Li, Prognostics and health management of lithium-ion battery using deep learning methods: A review, *Renewable and Sustainable Energy Reviews* 161 (2022) 112282.
- [41] B. Xu, A. Oudalov, A. Ulbig, G. Andersson, D. S. Kirschen, Modeling of lithium-ion battery degradation for cell life assessment, *IEEE Transactions on Smart Grid* 9 (2) (2016) 1131–1140.

- [42] W. He, N. Williard, M. Osterman, M. Pecht, Prognostics of lithium-ion batteries based on dempster–shafer theory and the bayesian monte carlo method, *Journal of Power Sources* 196 (23) (2011) 10314–10321.
- [43] C. Chen, M. Pecht, Prognostics of lithium-ion batteries using model-based and data-driven methods, in: *Proceedings of the IEEE 2012 Prognostics and System Health Management Conference (PHM-2012 Beijing)*, IEEE, 2012, pp. 1–6.
- [44] D. A. Najera-Flores, Z. Hu, M. Chadha, M. D. Todd, A physics-constrained bayesian neural network for battery remaining useful life prediction, *Applied Mathematical Modelling* (2023).
- [45] C. Lin, J. Xu, J. Hou, D. Jiang, Y. Liang, X. Zhang, E. Li, X. Mei, A fast data-driven battery capacity estimation method under non-constant current charging and variable temperature, *Energy Storage Materials* 63 (2023) 102967.
- [46] K. Li, P. Zhou, Y. Lu, X. Han, X. Li, Y. Zheng, Battery life estimation based on cloud data for electric vehicles, *Journal of Power Sources* 468 (2020) 228192.
- [47] N. Eleftheroglou, S. S. Mansouri, T. Loutas, P. Karvelis, G. Georgoulas, G. Nikolakopoulos, D. Zarouchas, Intelligent data-driven prognostic methodologies for the real-time remaining useful life until the end-of-discharge estimation of the lithium-polymer batteries of unmanned aerial vehicles with uncertainty quantification, *Applied Energy* 254 (2019) 113677.
- [48] J. Yang, C. Du, W. Liu, T. Wang, L. Yan, Y. Gao, X. Cheng, P. Zuo, Y. Ma, G. Yin, et al., State-of-health estimation for satellite batteries based on the actual operating parameters–health indicator extraction from the discharge curves and state estimation, *Journal of Energy Storage* 31 (2020) 101490.
- [49] D. Zhao, Z. Zhou, S. Tang, Y. Cao, J. Wang, P. Zhang, Y. Zhang, Online estimation of satellite lithium-ion battery capacity based on approximate belief rule base and hidden markov model, *Energy* 256 (2022) 124632.
- [50] G. Ma, S. Xu, B. Jiang, C. Cheng, X. Yang, Y. Shen, T. Yang, Y. Huang, H. Ding, Y. Yuan, Real-time personalized health status prediction of lithium-ion batteries using deep transfer learning, *Energy & Environmental Science* 15 (10) (2022) 4083–4094.

- [51] D. Roman, S. Saxena, V. Robu, M. Pecht, D. Flynn, Machine learning pipeline for battery state-of-health estimation, *Nature Machine Intelligence* 3 (5) (2021) 447–456.
- [52] N. Jiang, J. Zhang, W. Jiang, Y. Ren, J. Lin, E. Khoo, Z. Song, Driving behavior-guided battery health monitoring for electric vehicles using machine learning, arXiv preprint arXiv:2309.14125 (2023).
- [53] D. Zhang, Z. Wang, L. Peng, Z. Qin, Q. Wang, C. She, P. Bauer, Multi-step fast charging based state of health estimation of lithium-ion batteries, *IEEE Transactions on Transportation Electrification* (2023).
- [54] K. He, X. Zhang, S. Ren, J. Sun, Deep residual learning for image recognition, in: *Proceedings of the IEEE conference on computer vision and pattern recognition*, 2016, pp. 770–778.

REVIEWER COMMENTS

Reviewer #1 (Remarks to the Author):

The authors have added recent references that were not considered in the previous draft. However, after including references [39,40,41,42,43], they state, "There are few publications that implement battery degradation modeling or use machine learning methods to solve the battery degradation equation to achieve State of Health (SOH) estimation," which seems contradictory. The new paragraph added by the authors to justify the novelty of their study consists of three sentences:

"Although existing Physics-Informed Neural Networks (PINNs) have shown promising results in battery modeling and prognosis, most focus on modeling within one cycle (such as state-of-charge (SOC) estimation and end-of-discharge (EOD) prediction)."

I cannot agree with this statement, at least regarding the references where PINNs are used to model degradation, as this is not measured in a single cycle. Following this, it is stated that

"There are few publications that implement battery degradation modeling or use machine learning methods to solve the battery degradation equation to achieve SOH estimation."

Again, I consider this statement debatable, as there are numerous studies where machine learning is applied to estimate battery SOH, including a number of works with an electrochemical approach to degradation. Lastly, the authors indicate

"In this work, we propose a PINN for battery SOH estimation."

In the response letter to the reviewers, a categorization of methods defined in [36] is referenced, and a figure from reference [7] is included. On one hand, this explanation is not included in the paper. On the other, the justification in the letter still does not fully address my initial comment, which was, "It's challenging to pinpoint the exact contribution of the paper: it isn't the first to use PINNs for battery SOH modeling, and the SOH model used is simpler than others previously employed."

In the second comment, the authors state that the rest time after a CC-CV charge is not relevant since "whether it is fully discharged, it will eventually be fully charged." Whether a battery is considered "fully charged" or "fully discharged" depends on the experimental conditions and the procedure followed to interrupt the charge or discharge.

In the third comment, the authors are told that their paper is difficult to follow from start to finish and requires jumping back and forth several times. Their response is that the journal's outline forces them to organise their paper in this way. Given that other papers published in this same journal with the same outline do not suffer from this problem, I remain dissatisfied with the organisation of the paper.

Finally, the authors were informed that there were significant errors in the notation of the functions in section 4. The authors have redefined their variable $u(t)$ in equation 1, which is now a sequence $u(k)$, but as a result, their equation (2) now refers to $du(t)/dt$, which makes no sense if u is a discrete variable. I understand that they have a variable $u(t)$ and that in equation (1) they intended to define a sequence of values $u_k = u(t_k)$ for some temporal values t_k . The equation (2) in the revised version is now different from the one indicated in the initial draft; in the initial draft, it was a partial differential equation referring to a state variable x , which has been eliminated from the definition of the function $\{F\}$ in the revised draft. Subsequently, the authors "redefine" the variables as if they were writing a computer program; this is not an acceptable practice. The same equation is defined in various ways, making it difficult to ascertain whether $\{F\}$ depends on the variable x or not, whether u is a sequence or a continuous function. I also cannot make sense of the new equation 5, where a function is defined that,

according to equation (4), is $f := u_t - \{\text{cal F}\}(\text{etc}) = \frac{\{\partial u(t,x)\}}{\{\partial t\}} - \{\text{cal F}\}(\text{etc}) = \{\text{cal F}\}(\text{etc}) - \{\text{cal F}\}(\text{etc}) = 0$. The rest of the section contains errors of similar importance.

Reviewer #2 (Remarks to the Author):

I thank the authors for their reply and update.

I am still a bit concerned about the information leakage issue, as I think my question was not clear enough. Also, information leakage can fool experienced people, and I am not really qualified enough to spot these things from the manuscript.

My question about "what the prediction task really is" - does not refer to whether they are predicting SOH or RUL. What I meant was - are you trying to predict the variation in outcomes due to manufacturing variability? Are you trying to understand the outcomes based on design variability? Are you trying to predict outcomes, based on differences in charging time? Are you trying to predict the outcomes, based on different driving conditions? Leakage can occur, when using data/experiment that was designed to answer one of the above questions - but then that data is used to create a model to answer a different question.

The paper I referred to makes the point that if your data and prediction tasks don't align - you can trivially introduce the "answer" to the prediction task in the form of information leakage.

Again, leakage issues can be very subtle and I don't know that the authors have an issue. My concern lies in that the same features are being used across different datasets that were designed to test different scientific hypothesis - therefore my concern about leakage is from that simple observation alone.

I think the final version of the paper should include some text to acknowledge the problem and explain what steps the authors took to make sure the results are not "leaked". Again, I don't know of any test to claim there is no leakage - so I will have to trust that the authors have done this carefully.

Reviewer #3 (Remarks to the Author):

The manuscript was revised based on the comments and most questions were clarified in detail. In view of its low novelty value, the paper could nevertheless make an important contribution by explaining the implementation and application of a PINN in a clear manner and also discussing its limitations. However, the structure of the paper makes it difficult to understand and impedes readability. Unfortunately, this point of criticism was not addressed further. I therefore do not recommend publication in the present form.

Reviewer #3 (Remarks on code availability):

The code is well structured and readability is also good. However, it would be greatly appreciated if the comments in the code could be expanded and translated into English. Currently they are in Chinese.

Reviewer #4 (Remarks to the Author):

I co-reviewed this manuscript with one of the reviewers who provided the listed reports. This is part of the Nature Communications initiative to facilitate training in peer review and to provide appropriate recognition for Early Career Researchers who co-review manuscripts

Reviewer #4 (Remarks on code availability):

The attached code with it's description is comprehensive and understandable.

Response Letter

Journal Title: Nature Communications

Manuscript ID: NCOMMS-23-46244A

Title of Paper: Physics-informed neural network for lithium-ion battery degradation stable modeling and prognosis

Note: In this Response Letter, the **contents in blue** are the responses and discussions to the comments of editors and reviewers; the **contents in red** indicate that these contents have been added to the revised manuscript.

First of all, the authors would like to express their sincere thanks to the anonymous reviewers for their helpful comments and suggestions. The explanation of the modifications as well as corrections in this revision can be arranged as follows (comment numbers are in 1:1 correspondence with the reviewers' comments).

1 Response to Reviewer #1

Comment: The authors have added recent references that were not considered in the previous draft. However, after including references [39,40,41,42,43], they state, "There are few publications that implement battery degradation modeling or use machine learning methods to solve the battery degradation equation to achieve State of Health (SOH) estimation," which seems contradictory. The new paragraph added by the authors to justify the novelty of their study consists of three sentences:

"Although existing Physics-Informed Neural Networks (PINNs) have shown promising results in battery modeling and prognosis, most focus on modeling within one cycle (such as state-of-charge (SOC) estimation and end-of-discharge (EOD) prediction)."

I cannot agree with this statement, at least regarding the references where PINNs are used to model degradation, as this is not measured in a single cycle. Following this, it is stated that "There are few publications that implement battery degradation modeling or use machine learning methods to solve the battery degradation equation to achieve SOH estimation."

Again, I consider this statement debatable, as there are numerous studies where machine learning is applied to estimate battery SOH, including a number of works with an electro-chemical approach to degradation. Lastly, the authors indicate

"In this work, we propose a PINN for battery SOH estimation."

In the response letter to the reviewers, a categorization of methods defined in [36] is referenced, and a figure from reference [7] is included. On one hand, this explanation is not included in the paper. On the other, the justification in the letter still does not fully address my initial comment, which was, "It's challenging to pinpoint the exact contribution of the paper: it isn't the first to use PINNs for battery SOH modeling, and the SOH model used is simpler than others previously employed."

RESPONSE: Thank you again for reviewing our revised manuscript v1 and providing your valuable comments. We sincerely appreciate your feedback. We will respond to your questions and comments in points.

1. We are aware of the contradictions in our statements pointed out by the reviewer and apologize for them. First, we still give here the figure in reference [1] that appears in response letter v1, as shown in Figure 1. In this figure, Aykol et al. classified battery modeling methods that combine physical knowledge and machine learning into five categories, including three Sequential Integration methods, A1-A3, and two Hybrid methods, B1-B2. Among them, an obvious feature of the Sequential Integration method is that the physical model and the machine learning model are standalone, while the Hybrid method fuses the two together. The Sequential Integration method is relatively straightforward to implement because the physical

Figure 1: A screenshot of Fig. 1 from Aykol et al. [1].

model and the machine learning model are standalone, making it a practical near-term strategy for battery modeling. The Hybrid methods, on the other hand, are more fundamental as they truly integrate the primary governing equations for battery modeling with data-driven methods. However, due to the complex physical model (such as P2D model) contain numerous parameters and are difficult to solve, the Hybrid architectures for electrochemical modeling remains an open research question. In revised manuscript v1, what we want to state is that “there are not many publications using PINN (specifically the Hybrid approach, i.e. using physical equations to explicitly constrain the neural network or integrate physical equations into neural networks) to model the battery degradation process and estimate its SOH.” However, the lack of rigorous expression led to contradictions. To avoid such a contradiction, we have deleted this sentence in this revised manuscript and expressed it in a more appropriate sentence.

~~“ There are few publications that implement battery degradation modeling or use machine learning methods to solve the battery degradation equation to achieve SOH estimation.”~~

2. For the reviewer’s comment: The new paragraph added by the authors to justify the novelty of their study consists of three sentences: “Although existing ...” I cannot agree with this statement, at least regarding the references where PINNs are used to model degradation, as this is not measured in a single cycle. In our revised manuscript v1, we state: “Although existing PINNs have shown promising results in battery modeling and prognosis, most focus on modeling within one cycle (such as state-of-charge (SOC) estimation and end-of-discharge (EOD) prediction).” We understand your skepticism about this view, as there does exist some literature using PINNs to model battery degradation that is not limited to a single cycle. We sincerely appreciate you pointing this out. To avoid unnecessary misunderstandings and ambiguities, we have deleted this sentence in this revised manuscript.

~~“ Although existing PINNs have shown promising results in battery modeling and prognosis, most of them focus on modeling within one cycle (such as state-of-charge (SOC) estimation and end-of-discharge (EOD) prediction). ”~~

3. For the reviewer’s comment: Following this, it is stated that “There are few ... ” Again, I consider this statement debatable, as there are numerous studies where machine learning is applied to estimate battery SOH, including a number of works with an electrochemical approach to degradation. Lastly, the authors indicate “In this work, we propose a PINN for battery SOH estimation.” We apologize for the loose statement in revised manuscript v1. We understand your doubts about this point of view and thank you again for your valuable comments. This is caused by our improper expression. What we want to express is that there are not many publications **using PINN (specifically the Hybrid approach, i.e. using physical equations to explicitly constrain the neural network or integrate physical**

equations into neural networks), rather than machine learning, to model the battery degradation process and estimate its SOH. As the reviewer said, there is a large amount of research applying machine learning methods to estimate the SOH of batteries, including a number of works with an electrochemical methods. In contrast, there are few works using PINN (specifically the Hybrid approach in [1]) to model battery SOH, and it is worthy of further exploration by scholars. In our revised manuscript v1, we wanted to use this sentence to introduce the contribution of our work, but it caused ambiguity due to improper expression. In this revised manuscript, we have deleted this sentence and modified it to include other words to express the difference between our work and existing publications and our contributions.

4. For the reviewer’s comment: In the response letter to the reviewers, a categorization of methods defined in [1] is referenced, and a figure from reference [1] is included. On one hand, this explanation is not included in the paper. On the other, the justification in the letter still does not fully address my initial comment, which was, ”It’s challenging to pinpoint the exact contribution of the paper: it isn’t the first to use PINNs for battery SOH modeling, and the SOH model used is simpler than others previously employed .” We further clarify our statement in response to reviewer comment. However, PINN is a broad definition and it can have multiple architectures types and implementations, as classified in [1] (Figure 1). According to our literature survey and understanding, most existing PINNs for battery SOH modeling belong to the Sequential Integration method in [1], and we have also classified the methods reviewed in this revised manuscript. These methods utilize physics-based models (such as P2D, SPM, ECM, etc.) for data argumentation, or use ML to learn the parameters of the physics-based model, so their physical model part is more complex. However, more complex models do not necessarily mean better. Our method belongs to category B2, and the implementation of the method is different from existing methods. At the same time, we also demonstrate the superiority of the proposed method in small sample learning and transfer learning among batteries with different chemistries and charging/discharging profiles. These are the ways in which our method differs from existing methods, and the contributions of our work. In this revised manuscript, we have added the classification and explanations in reference [36] to the article based on the reviewer’ comments, and stated the differences between our method and those in other publications in detailed. **To summarize, we’ve listed all the changes made to this comment below.**

“Aykol et al. [1] classified battery modeling methods that combine physical knowledge and machine learning into five categories, including three Sequential Integration methods, A1-A3, and two Hybrid methods, B1-B2. Among them, an obvious feature of the Sequential Integration method is that the physical model and the machine learning model are standalone, while the Hybrid method fuses the two together.

According to the architectures proposed by Aykol et al. [1], these methods belong to the A2 [2; 3; 4] and A3 [5; 6] categories. ”

“In fact, the Sequential Integration method is relatively straightforward to implement because the physical model and the machine learning model are standalone, making it a practical near-term strategy for battery modeling. Essentially, machine learning models in Sequential Integration method are not subject to physical constraints. The Hybrid methods, on the other hand, are more fundamental as they truly integrate the primary governing equations for battery modeling with data-driven methods. However, due to the complex physical equations contain numerous parameters and are difficult to solve, there are few publications that implement Hybrid methods for SOH estimation. Recent review [1] pointed out that Hybrid methods will become the dominant method in the long term, but it is still an open research question.”

“In this work, we proposed a PINN for battery SOH estimation, which belong to the B2 architecture. This approach achieves true integration of governing equations and neural networks, resulting in stable and precise SOH estimation. Unlike existing PINN approaches, we also validated its advancements in small sample learning and transfer learning among batteries with different chemistries and charge/discharge profiles. Specifically, first, considering the complexity of the electrochemical equations, it hinders the development of B2-type PINNs. Therefore, we model battery degradation dynamics from the perspective of empirical degradation and state space equations, and utilize neural networks to capture battery degradation dynamics. ...”

Comment: In the second comment, the authors state that the rest time after a CC-CV charge is not relevant since “whether it is fully discharged, it will eventually be fully charged.” Whether a battery is considered ”fully charged” or ”fully discharged” depends on the experimental conditions and the procedure followed to interrupt the charge or discharge.

RESPONSE: Thanks for reviewer’s comment. We agree with the reviewer that whether a battery is considered “fully charged” or “fully discharged” depends on the experimental conditions and the procedures followed to interrupt the charge or discharge. We apologize for the misunderstanding caused by the unclear reply in revised manuscript v1. In Revision 1, what we want to state is that the rest time after CC-CV charging is *not important for our proposed SOH estimation method* because our method does not use the data during the rest period. In addition, our statement “whether it is fully discharged, it will eventually be fully charged” is for the four datasets we used, and it is a description of the datasets we used. We understand the doubts raised by the reviewer. In this revision, we have modified the relevant statements to avoid misunderstanding and confusion.

“... For the four public datasets we used, no matter what strategy the battery is discharged with or whether it is fully discharged, it will eventually be fully charged when charging.”

Comment: In the third comment, the authors are told that their paper is difficult to follow from start to finish and requires jumping back and forth several times. Their response is that the journal’s outline forces them to organise their paper in this way. Given that other papers published in this same journal with the same outline do not suffer from this problem, I remain dissatisfied with the organisation of the paper.

RESPONSE: We sincerely appreciate your comments and would like to explain further. We take your questions about the organization of the article very seriously. In our response letter v1, we mentioned that the journal’s outline requirements require us to organize the content of the article in a certain order, and this point of view does exist. However, we understand your concerns and are aware that similar issues have not arisen in other articles published in the same journal. To solve this problem, we have **reorganized** the content of Section *2.1 Framework overview and flowchart* and also redrawn **Figure 1** to make it more coherent and easier to understand. Specifically, in **Figure 1**, we draw the complete flowchart of the proposed PINN for SOH estimation. We also refine the architecture of PINN and add some formulas in **Figure 1(c)**. Combined with the text description in Section 2.1, readers can first have a preliminary understanding of our method. On this basis, they can smoothly read and understand the paper from the beginning to the end, avoiding the need for readers to jump back and forth. When the reader reads the **Method** section, he/she can learn in more detail about our ideas when proposing the PINN and the analysis and implementation of the PINN, so as to achieve a deeper understanding of the process of the PINN after knowing the results of that. The revised content of *2.1 Framework overview and flowchart* and **Figure 1** are as follows. We also hope that these revisions can effectively resolve your concerns and help improve your reading experience.

2.1 Framework overview and flowchart

We developed a PINN for lithium-ion battery SOH estimation, and its flowchart is shown in **Figure 2**. Our method is designed for more general, reliable, stable, and high-precision SOH estimation by considering the dynamic behavior of battery degradation as well as the degradation trend.

In the data preprocessing stage (shown in **Figure 2(b)**), statistical features are extracted from a short period of data before the battery is fully charged as the input of the model, which ensures that this period of data exists in most battery datasets, and solves the problem of non-universal features in existing studies. Therefore, our method is applicable to batteries with different chemistries and charge/discharge protocols.

Figure 2: The flowchart of the proposed PINN for lithium-ion battery SOH estimation. (a). The lithium-ion batteries may have different chemistries. Different users have personalized battery discharge strategies resulting in different degradation trajectories. (b). An illustration of the selected data for feature extraction. We extracted features from a short period of data before the battery is fully charged. These features are used as the inputs of the proposed PINN to estimate SOH. (c). The structure of the proposed PINN (see section 4 for more details).

In the SOH estimation stage, due to the complexity of electrochemical equations, there is currently no good way to integrate them with the neural networks. In this work, we modeled the attributes that affect the battery degradation from the perspective of the empirical degradation and state space equation, and utilized neural networks to approximate the established degradation model, effectively achieving the integration of governing equations and neural networks. The proposed PINN consists of two parts: a solution function $f(\cdot)$ that maps features to SOH and a nonlinear function $g(\cdot)$ that models battery degradation dynamic behaviors, as shown in the Figure 2(c). The solution $f(\cdot)$, modeling the relationship between features and SOH, is expressed as $u^i = f(t^i, \mathbf{x}^i)$, where t^i represents time, \mathbf{x}^i represents the extracted feature vector, and u^i denotes the SOH of the cycle i . The nonlinear function $g(\cdot)$ models the SOH decay rate of the battery. Since $f(\cdot)$ and $g(\cdot)$ are affected by many factors in reality and their explicit expressions are unknown, they are replaced by small fully connected neural networks, denote as $\mathcal{F}(\cdot)$ and $\mathcal{G}(\cdot)$. During training, we consider data term loss $\mathcal{L}_{\text{data}}$, monotonicity loss $\mathcal{L}_{\text{mono}}$, and loss \mathcal{L}_{PDE} constrained by the degradation equation described by partial differential equation. They minimize the errors between the predicted and the true values, while making the model follow the properties of monotonicity of the degradation trajectory and satisfy the constraints of the established degradation model.

To validate the superiority of the proposed PINN, we conducted small sample experiments and transfer experiments. During the transfer experiments, we froze $\mathcal{G}(\cdot)$ and fine-tuned $\mathcal{F}(\cdot)$ on datasets with different chemical compositions. The experimental outcomes demonstrate that the proposed PINN framework can effectively capture the dynamics of battery degradation. Our study combines knowledge of the battery degradation with neural networks and achieves promising results. This study highlights the promise of physics-informed neural network for battery degradation modeling and SOH estimation (more methodological details can be found in Section 4).

Comment: Finally, the authors were informed that there were significant errors in the notation of the functions in section 4. The authors have redefined their variable $u(t)$ in equation 1, which is now a sequence $u(k)$, but as a result, their equation (2) now refers to $du(t)/dt$, which makes no sense if u is a discrete variable. I understand that they have a variable $u(t)$ and that in equation (1) they intended to define a sequence of values $u_k = u(t_k)$ for some temporal values t_k . The equation (2) in the revised version is now different from the one indicated in the initial draft; in the initial draft, it was a partial differential equation referring to a state variable x , which has been eliminated from the definition of the function \mathcal{F} in the revised draft. Subsequently, the authors "redefine" the variables as if they were writing a computer program; this is not an acceptable practice. The same equation is defined in various ways, making it

difficult to ascertain whether \mathcal{F} depends on the variable x or not, whether u is a sequence or a continuous function. I also cannot make sense of the new equation 5, where a function is defined that, according to equation (4), is $f := u_t - \mathcal{F}(\dots) = \frac{\partial u(t, \mathbf{x})}{\partial t} - \mathcal{F}(\dots) = \mathcal{F}(\dots) - \mathcal{F}(\dots) = 0$. The rest of the section contains errors of similar importance.

RESPONSE: Thank you for reviewing our article again and pointing out a errors regarding function symbols in Section 4. We attach great importance to your questions and will provide detailed responses and modifications.

We do intend to define a sequence of values $u_k = u(t_k)$ to represent the variable values at a series of time points t_k . However, we also realize that due to this change in variable definition, the reference to $du(t)/dt$ in equation (2) is no longer appropriate since u is now a discrete variable. We are indeed aware of this error and have corrected it in the revised manuscript.

In addition, regarding your doubts about equation 5 ($f := u_t - \mathcal{F}(t, \mathbf{x}, u; \Theta)$), we realize that we did not express it clearly. Let us explain this: The definition of $f := u_t - \mathcal{F}(t, \mathbf{x}, u; \Theta)$ is the standard definition of PINN, originally from the article *Physics-informed neural networks: A deep learning framework for solving forward and inverse problems involving nonlinear partial differential equation*. If the explicit forms of $u(\cdot)$ and $\mathcal{F}(\cdot)$ in the equation are known, then as to the reviewer’s understands, f defined in equation 5 is strictly equal to 0. However, in our method, since the internal degradation mechanism of the battery is not known, both $u(\cdot)$ and $\mathcal{F}(\cdot)$ are fitted with neural networks, so f is not strictly equal to 0. The optimization objective of the model is to make f approach to 0, so that the battery degradation mechanism learned by the model satisfies the constraints of equation 4 ($\frac{\partial u(t, \mathbf{x})}{\partial t} = \mathcal{F}(t, \mathbf{x}, u; \Theta)$).

In this revised manuscript, we realized the problem of unclear formula description and redefinition, so we **reorganized and rewritten** the entire **Method** section. It is worth noting that **the symbols have also been completely replaced**. For the sake of uniformity, we use a single lowercase letter to represent variables, such as u ; a lowercase letter followed by parentheses to represent a function, such as $f(\cdot)$; and an uppercase “mathcal” font to represent a neural network, such as $\mathcal{F}(\cdot)$.

4.1 Battery degradation modeling

Battery aging is primarily characterized by a decrease in available capacity and an increase in internal resistance, typically following a declining trajectory. To accurately describe the battery degradation trajectory, scholars have proposed various empirical models to describe the loss of battery capacity as a function of time or cycle numbers, including the linear model [7], exponential model [8; 9], power-law model [10], and failure forecast model (FFM) [11], etc. These models all describe the battery’s degradation trajectory as a univariate function of time.

However, representing the degradation trajectory of batteries solely as a univariate function

of time oversimplifies the process. In fact, battery degradation is not only related to time but also related to charging rate, discharging rate, calendar time, temperature, depth of discharge (DOD), etc. For example, Xu et al. [12] divided battery aging into calendar aging and cycle aging, which considered factors such as state-of-charge (SOC), DOD, cell temperature, and solid electrolyte interphase (SEI) film growth. They modeled calendar aging and cycle aging as functions of calendar time, SOC, DOD, and temperature.

Therefore, modeling the degradation trajectory of a battery solely as a function of time is inadequate. In this study, we propose to model it as a multivariate function:

$$u = f(t, \mathbf{x}), \quad (1)$$

where t represents time and \mathbf{x} represents a vector composed of SOC, DOD, temperature, charge rate, discharge rate, health indicators (HIs), and all other factors. In our work, \mathbf{x} represents the HIs extracted from the charging data (see 2.3 for more details).

Without loss of generality, to describe the degradation dynamics of the battery, its SOH decay rate can be described as:

$$\frac{\partial u}{\partial t} = g(t, \mathbf{x}, u; \theta). \quad (2)$$

The above equation is an explicit partial differential equation (PDE) parameterized by θ , and $g(\cdot)$ represents the nonlinear function of t , \mathbf{x} , and u . The function $g(\cdot)$ characterizes the internal degradation dynamics of the battery, and by altering this nonlinear function, various forms of degradation can be represented. Models such as linear model, exponential model, power-law model, and FFM can be viewed as particular cases of Equation (2) when only the time is considered.

4.2 Physics-informed neural network

An unavoidable problem is that the explicit form of $g(\cdot)$ is unknown and difficult to obtain. In response to similar problems, Sun et al. [13] proposed a sparse regression physics-informed neural network that exploits sparsity to learn the parameters θ of $g(\cdot)$ from a given candidate set. Raissi et al. [14] proposed deep hidden physics models to model $g(\cdot)$. Inspired by [14; 15], we propose to use a more generalized function approximator $g'(\cdot)$ with parameters θ' to represent the nonlinear dynamics $g(\cdot)$. Therefore, Equation (2) becomes:

$$u_t \approx g' \left(t, \mathbf{x}, u, u_t, u_{\mathbf{x}}, u_{\mathbf{x}\mathbf{x}}, \dots; \theta' \right). \quad (3)$$

In the equation, $u_t = \frac{\partial u}{\partial t}$, we employ a neural network $\mathcal{F}(t, \mathbf{x}; \Phi)$ with learnable parameters Φ to model $f(t, \mathbf{x})$ and utilize automatic differentiation mechanisms to compute u_t . $u_{\mathbf{x}} = \left[\frac{\partial u}{\partial x_1}, \frac{\partial u}{\partial x_2}, \dots \right]^\top$ represents the first-order partial derivative of u with respect to \mathbf{x} , and $u_{\mathbf{x}\mathbf{x}}$

represents the second-order partial derivative. One advantage of this approach is that we do not need to specify a candidate basis function set for $g(\cdot)$, but instead employ a more generalized approximators $g'(\cdot)$. The function approximator $g'(\cdot)$ propose a more flexible relationship to t , u , \mathbf{x} , and their arbitrary order partial derivatives. A neural network $\mathcal{G}(\cdot)$ with learnable parameters Θ is used to model $g'(\cdot)$ so that it can learn the aging mechanism of the battery from the given \mathbf{x} , t , and other partial derivatives.. To balance accuracy and computational complexity, we only consider the influence of first-order partial derivatives, discarding higher-order derivatives.

Building upon the aforementioned analysis, we define a physics-informed neural network \mathcal{H} [16; 15] for battery aging:

$$\mathcal{H} := \frac{\partial \mathcal{F}(t, \mathbf{x}; \Phi)}{\partial t} - \mathcal{G}(t, \mathbf{x}, u, u_t, u_{\mathbf{x}}; \Theta), \quad (4)$$

where $\frac{\partial \mathcal{F}(t, \mathbf{x}; \Phi)}{\partial t}$ represents the partial derivation of solution neural network $\mathcal{F}(\cdot)$ with respect to t , and $\mathcal{G}(\cdot)$ denotes the battery degradation dynamic equation modeled by the neural network. The structure of the proposed PINN is shown in Figure 3.

Equation (4) is derived from Equation (2) and Equation (3). However, since it is fitted by a neural network, its training process is discrete, so it does not strictly satisfy Equation (2). For battery SOH, the calculation formula is [17]:

$$u^k = f(k, \mathbf{x}) = \frac{Q^k}{Q^0}, \quad (5)$$

where Q^k represents the capacity of cycle k and Q^0 represents the nominal capacity. The SOH value u^k coincides with the point on the degradation trajectory $f(\cdot)$ when $t = k$. We need to make $\mathcal{H}(t^i, \mathbf{x}^i) = 0$ hold at sample point i to approximate Equation (2). Therefore, the optimization process of the PINN needs to adhere to the PDE loss specified by Equation (2), i.e.:

$$\mathcal{L}_{\text{PDE}} = \sum_{i=1}^N |\mathcal{H}(t^i, \mathbf{x}^i)|^2, \quad (6)$$

where superscript i denotes the i -th sample and N denotes the number of samples. Also, the optimization objective includes data item loss and monotonicity loss:

$$\mathcal{L}_{\text{data}} = \sum_{i=1}^N |u^i - \hat{u}^i|^2, \quad (7)$$

$$\mathcal{L}_{\text{mono}} = \sum_{j=1}^M \sum_{k=1}^{N_j} \text{ReLU}(\hat{u}^{k+1} - \hat{u}^k), \quad (8)$$

where \hat{u}^t represents the estimated SOH, M represents the number of batteries, N_j denotes the number of cycles of battery j , and $\text{ReLU}(\cdot)$ is Rectified Linear Unit. The monotonicity loss $\mathcal{L}_{\text{mono}}$ is based on the physical properties of battery degradation, that is, the SOH of the next cycle should be less than or equal to that of the previous cycle (unless capacity regeneration occurs). The total function is formulated as:

$$\mathcal{L} = \mathcal{L}_{\text{data}} + \alpha \mathcal{L}_{\text{PDE}} + \beta \mathcal{L}_{\text{mono}}, \quad (9)$$

where the α and β are trade-off parameters. More details about our model can be found in Supplementary Note 3.

Figure 3: An illustration of the proposed physics-informed neural network.

4.3 Transfer learning with physics-informed neural network

Our PINN for battery aging consists of two parts: a solution neural network $\mathcal{F}(\cdot)$ that builds the feature-to-SOH mapping and a neural network $\mathcal{G}(\cdot)$ that models battery degradation dynamics, as shown in Figure 3. We believe that the degradation dynamics $\mathcal{G}(\cdot)$ are independent of charge/discharge protocols and datasets, while the solution $\mathcal{F}(\cdot)$ is related to them. Therefore, when our PINN is applied to transfer learning scenarios, dynamics $\mathcal{G}(\cdot)$ is frozen, and only solution $\mathcal{F}(\cdot)$ is fine-tuned, as shown in Figure 3(b).

2 Response to Reviewer #2

Comment: I thank the authors for their reply and update.

I am still a bit concerned about the information leakage issue, as I think my question was not clear enough. Also, information leakage can fool experienced people, and I am not really qualified enough to spot these things from the manuscript.

My question about "what the prediction task really is" - does not refer to whether they are predicting SOH or RUL. What I meant was - are you trying to predict the variation in outcomes due to manufacturing variability? Are you trying to understand the outcomes based on design variability? Are you trying to predict outcomes, based on differences in charging time? Are you trying to predict the outcomes, based on different driving conditions? Leakage can occur, when using data/experiment that was designed to answer one of the above questions - but then that data is used to create a model to answer a different question.

The paper I referred to makes the point that if your data and prediction tasks don't align - you can trivially introduce the "answer" to the prediction task in the form of information leakage.

Again, leakage issues can be very subtle and I don't know that the authors have an issue. My concern lies in that the same features are being used across different datasets that were designed to test different scientific hypothesis - therefore my concern about leakage is from that simple observation alone.

I think the final version of the paper should include some text to acknowledge the problem and explain what steps the authors took to make sure the results are not "leaked". Again, I don't know of any test to claim there is no leakage - so I will have to trust that the authors have done this carefully.

RESPONSE: Thank you for your important comment about our manuscript. We take your concerns about information leakage very seriously and are open to further discussions and revisions on this version.

First of all, we deeply apologize as we may not have clearly understood your initial question about the prediction task. Before this revision, we have looked for some more information about "data leakage" to learn, and based on that, we try to answer the questions you have asked.

In response to your mention of "SOH depends on both battery internal quality and usage conditions", as well as the inquiry "are you trying to predict...", let us provide clarification. Our model's inputs do not include internal quality information or usage conditions. In other words, we do not utilize data such as manufacturing variability, design variability, driving conditions, etc., to predict variation in outcomes. Charging time is indeed included as a feature

in the inputs to our model. However, it denotes the time span of the data segment we have intercepted, which can reflect information about the battery degradation process.

The reviewer’s concern lies in that the same features are being used across different datasets that were designed to test different scientific hypothesis. We endeavor to address this concern.

The selection of these features stems from the fact that variations in the charging curves of batteries can reflect their aging process, and we have extracted these statistical features from partial charging curves. As batteries undergo aging, various interface degradation processes occur, along with the loss of lithium inventory and active materials, leading to increased resistance in ion and electron transfer as well as intercalation reactions, thereby resulting in changes in their charging curves [18]. Consequently, charging curves harbor rich degradation process information. The segment of data we chose before the battery was fully charged can be seen as a measure of the local information (utility) of the charging curve, which varies in relation to energy dissipation and internal resistance accumulation [19]. In fact, we are not the only ones who have done this to extract features from charging (or discharging) curves. There are a number of published articles that have extracted features from charging/discharging curves as inputs to their models [20; 21; 22].

Based on the concerns raised by the reviewer, we checked some information about “information/data leakage” and found some traps of “information/data leakage”:

1. Improper handling of time series data: when handling time series data, the data in the previous is used to predict the future data, and the future data is accidentally leaked into the training set, resulting in data leakage. In our approach, we train our model with data from battery A and test it on battery B; instead of training the model with early data from battery A and testing it with later data, which avoids information leakage from the training set to the test set. Therefore, all the features extracted are not leaked during the training process.
2. The use of features with future information: e.g., labels or metrics generated based on future events, can lead to leakage. Our method does not involve this issue.
3. Incorrect cross-validation: during cross-validation, if the data in the test set is “seen” by the model during the training phase, this can also result in data leakage. Our model does not use the cross-validation method, and the problem of the model having seen the test data during the training phase does not occur.
4. Model leakage: Leaks can occur when model outputs or predictions are incorrectly used as new training data or features. There are no such errors in our approach either.

Relevant information shows that if the model performs too perfectly on the test set, it is likely

that a data leak has occurred. From Table 4 in the manuscript, “Source-only” results are worse when we conduct transfer experiments between different datasets. Therefore, this also indicates that there is no data leakage in our operation.

The reviewer would like us to add a paragraph to the manuscript acknowledging the problem and explaining what steps we took to ensure that the results were not “leaked”. In this revision, we added relevant statements in Section 3 **Discussion**:

“Battery degradation modeling and SOH estimation are research hotspots in the field of battery health management. As batteries aging, various interface degradation processes occur, along with the loss of lithium inventory and active materials, leading to increased resistance in ion and electron transfer as well as intercalation reactions, thereby resulting in changes in their charging curves [18]. Consequently, the charging curve contains rich information on the degradation process. **However, using charge and discharge curves to estimate battery SOH may fall into the trap of information leakage. Geslin et al. [23] pointed out that inconsistent charging and discharging protocols, usage conditions, etc. may lead to information leakage, which is a serious problem that may be ignored by scholars.** They believe that a fixed CC-CV mode can alleviate the problem of information leakage. Hence, it is advisable to avoid incorporating factors related to internal battery quality, manufacturing variability, and usage conditions as much as possible when performing SOH estimation tasks. In our study, the features are extracted from a small segment of data from the CC-CV stage before the battery is fully charged, which is independent of the battery usage conditions. This ensures the usefulness and versatility of the features we extracted, while avoiding the problem of information leakage caused by inconsistent charging protocols or battery usage conditions. During the training and test stage, we train the model with data from battery A and test it on battery B; instead of training the model with early data from battery A and testing it with later data, which avoids information leakage from the training set to the test set. ”

3 Response to Reviewer #3

Comment: The manuscript was revised based on the comments and most questions were clarified in detail.

In view of its low novelty value, the paper could nevertheless make an important contribution by explaining the implementation and application of a PINN in a clear manner and also discussing its limitations. However, the structure of the paper makes it difficult to understand and impedes readability. Unfortunately, this point of criticism was not addressed further.

I therefore do not recommend publication in the present form.

RESPONSE: Thank you for your review of our revised manuscript v1, and we are sorry that revision v1 did not address your concerns. We attach great importance to your concerns about the structure and readability of the article and will provide further explanations and revisions in this revision.

In response to the issues mentioned by the reviewer that the structure of the paper makes it difficult to understand and impedes readability, we have **reorganized** the content of Section *2.1 Framework overview and flowchart* and also redrawn **Figure 1** to make it more coherent and easier to understand. Specifically, in **Figure 1**, we draw the complete flowchart of the proposed PINN for SOH estimation. We also refine the architecture of PINN and add some formulas in **Figure 1(c)**. Combined with the text description in Section 2.1, readers can first have a preliminary understanding of our method. On this basis, they can smoothly read and understand the paper from the beginning to the end, avoiding the need for readers to jump back and forth. When the reader reads the **Method** section, he/she can learn in more detail about our ideas when proposing the PINN and the analysis and implementation of the PINN, so as to achieve a deeper understanding of the process of the PINN after knowing the results of that. The revised content of *2.1 Framework overview and flowchart* and **Figure 1** are as follows. We also hope that these revisions can effectively resolve your concerns and help improve your reading experience.

2.1 Framework overview and flowchart

We developed a PINN for lithium-ion battery SOH estimation, and its flowchart is shown in **Figure 2**. Our method is designed for more general, reliable, stable, and high-precision SOH estimation by considering the dynamic behavior of battery degradation as well as the degradation trend.

In the data preprocessing stage (shown in **Figure 2(b)**), statistical features are extracted from a short period of data before the battery is fully charged as the input of the model, which ensures that this period of data exists in most battery datasets, and solves the problem of non-universal features in existing studies. Therefore, our method is applicable to batteries

Figure 4: The flowchart of the proposed PINN for lithium-ion battery SOH estimation. (a). The lithium-ion batteries may have different chemistries. Different users have personalized battery discharge strategies resulting in different degradation trajectories. (b). An illustration of the selected data for feature extraction. We extracted features from a short period of data before the battery is fully charged. These features are used as the inputs of the proposed PINN to estimate SOH. (c). The structure of the proposed PINN (see section 4 for more details).

with different chemistries and charge/discharge protocols.

In the SOH estimation stage, due to the complexity of electrochemical equations, there is currently no good way to integrate them with the neural networks. In this work, we modeled the attributes that affect the battery degradation from the perspective of the empirical degradation and state space equation, and utilized neural networks to approximate the established degradation model, effectively achieving the integration of governing equations and neural networks. The proposed PINN consists of two parts: a solution function $f(\cdot)$ that maps features to SOH and a nonlinear function $g(\cdot)$ that models battery degradation dynamic behaviors, as shown in the Figure 4(c). The solution $f(\cdot)$, modeling the relationship between features and SOH, is expressed as $u^i = f(t^i, \mathbf{x}^i)$, where t^i represents time, \mathbf{x}^i represents the extracted feature vector, and u^i denotes the SOH of the cycle i . The nonlinear function $g(\cdot)$ models the SOH decay rate of the battery. Since $f(\cdot)$ and $g(\cdot)$ are affected by many factors in reality and their explicit expressions are unknown, they are replaced by small fully connected neural networks, denote as $\mathcal{F}(\cdot)$ and $\mathcal{G}(\cdot)$. During training, we consider data term loss $\mathcal{L}_{\text{data}}$, monotonicity loss $\mathcal{L}_{\text{mono}}$, and loss \mathcal{L}_{PDE} constrained by the degradation equation described by partial differential equation. They minimize the errors between the predicted and the true values, while making the model follow the properties of monotonicity of the degradation trajectory and satisfy the constraints of the established degradation model.

To validate the superiority of the proposed PINN, we conducted small sample experiments and transfer experiments. During the transfer experiments, we froze $\mathcal{G}(\cdot)$ and fine-tuned $\mathcal{F}(\cdot)$ on datasets with different chemical compositions. The experimental outcomes demonstrate that the proposed PINN framework can effectively capture the dynamics of battery degradation. Our study combines knowledge of the battery degradation with neural networks and achieves promising results. This study highlights the promise of physics-informed neural network for battery degradation modeling and SOH estimation (more methodological details can be found in Section 4).

In addition, we value the reviewer’s comment that “the paper could nevertheless make an important contribution by explaining the implementation and application of a PINN in a clear manner and also discussing its limitations”. Therefore, we have added more discussion of PINN in Section 3. **Discussion.**

“When building the SOH estimation model, we proposed a PINN for battery SOH estimation. Physics-informed neural network holds promise as an effective avenue for leveraging artificial intelligence to address practical engineering problems. By amalgamating traditional physics models with neural network models, it can more accurately capture the intricate dynamic behavior of battery systems, thereby facilitating more reliable and precise state estimation. However, this burgeoning field still requires further exploration by scholars. Within

the framework proposed by Aykol et al. [1], Hybrid methods, which utilize physical equations to constrain neural networks or integrate physical equations into neural networks, will become dominant in the long term. This class of hybrid methods have the potential to blend the causality and extrapolation capabilities of physics-based models with the speed, flexibility, and high-dimensional capabilities of neural networks. However, the limitation of these methods lies in the complexity of the battery’s physical model (e.g., the P2D model), which contains numerous parameters, and the internal parameters of the battery are difficult to collect. There is currently no satisfactory method to seamlessly integrate physical models and neural networks. The PINN proposed in this paper is modeled from the perspective of empirical degradation and state space equations, serving merely as an exploration of such hybrid methods and acting as a catalyst for further research. Additionally, we only consider extracting features from easily accessible current and voltage data. As more data and internal variables become available, more complex electrochemical models can be considered. The optimal integration of battery governing equations and neural networks for health management within the constraints of existing data and computational resources remains ripe for further exploration. ”

Comment: (Remarks on code availability) The code is well structured and readability is also good. However, it would be greatly appreciated if the comments in the code could be expanded and translated into English. Currently they are in Chinese.

RESPONSE: Thanks to the reviewer for your positive comments on our code. In this revision, we have added English translations after the Chinese comments based on the reviewer’s suggestions to facilitate reading by scholars from various countries. Please view our source code for modifications. We will also continue to improve our code for public release.

4 Response to Reviewer #4

Comment: I co-reviewed this manuscript with one of the reviewers who provided the listed reports. This is part of the Nature Communications initiative to facilitate training in peer review and to provide appropriate recognition for Early Career Researchers who co-review manuscripts

RESPONSE: Thank you for your valuable contribution as a co-reviewer in evaluating our manuscript. We highly value the insights and perspectives shared by both you and the other reviewers. Your comments have significantly contributed to the enhancement of our manuscript. Once again, we extend our gratitude for your contribution and constructive feedback, which will undoubtedly help us in refining our manuscript to meet the required standards for publication.

Comment: (Remarks on code availability) The attached code with it's description is comprehensive and understandable.

RESPONSE: Thanks to the reviewer for your positive comments on our code. We will also continue to improve our code for public release.

References

- [1] M. Aykol, C. B. Gopal, A. Anapolsky, P. K. Herring, B. van Vlijmen, M. D. Berliner, M. Z. Bazant, R. D. Braatz, W. C. Chueh, B. D. Storey, Perspective—combining physics and machine learning to predict battery lifetime, *Journal of The Electrochemical Society* 168 (3) (2021) 030525.
- [2] A. Thelen, Y. H. Lui, S. Shen, S. Laflamme, S. Hu, H. Ye, C. Hu, Integrating physics-based modeling and machine learning for degradation diagnostics of lithium-ion batteries, *Energy Storage Materials* 50 (2022) 668–695.
- [3] J. Shi, A. Rivera, D. Wu, Battery health management using physics-informed machine learning: Online degradation modeling and remaining useful life prediction, *Mechanical Systems and Signal Processing* 179 (2022) 109347.
- [4] T. Hofmann, J. Hamar, M. Rogge, C. Zoerr, S. Erhard, J. P. Schmidt, Physics-informed neural networks for state of health estimation in lithium-ion batteries, *Journal of The Electrochemical Society* 170 (9) (2023) 090524.
- [5] R. G. Nascimento, F. A. Viana, M. Corbetta, C. S. Kulkarni, A framework for li-ion battery prognosis based on hybrid bayesian physics-informed neural networks, *Scientific Reports* 13 (1) (2023) 13856.
- [6] F. Wang, Q. Zhi, Z. Zhao, Z. Zhai, Y. Liu, H. Xi, S. Wang, X. Chen, Inherently interpretable physics-informed neural network for battery modeling and prognosis, *IEEE Transactions on Neural Networks and Learning Systems* (2023) 1–15 [doi:https://doi.org/10.1109/TNNLS.2023.3329368](https://doi.org/10.1109/TNNLS.2023.3329368).
- [7] R. Spotnitz, Simulation of capacity fade in lithium-ion batteries, *Journal of power sources* 113 (1) (2003) 72–80.
- [8] W. He, N. Williard, M. Osterman, M. Pecht, Prognostics of lithium-ion batteries based on dempster–shafer theory and the bayesian monte carlo method, *Journal of Power Sources* 196 (23) (2011) 10314–10321.
- [9] C. Chen, M. Pecht, Prognostics of lithium-ion batteries using model-based and data-driven methods, in: *Proceedings of the IEEE 2012 Prognostics and System Health Management Conference (PHM-2012 Beijing)*, IEEE, 2012, pp. 1–6.

- [10] V. Ramadesigan, K. Chen, N. A. Burns, V. Boovaragavan, R. D. Braatz, V. R. Subramanian, Parameter estimation and capacity fade analysis of lithium-ion batteries using reformulated models, *Journal of the electrochemical society* 158 (9) (2011) A1048.
- [11] D. A. Najera-Flores, Z. Hu, M. Chadha, M. D. Todd, A physics-constrained bayesian neural network for battery remaining useful life prediction, *Applied Mathematical Modelling* (2023).
- [12] B. Xu, A. Oudalov, A. Ulbig, G. Andersson, D. S. Kirschen, Modeling of lithium-ion battery degradation for cell life assessment, *IEEE Transactions on Smart Grid* 9 (2) (2016) 1131–1140.
- [13] Z. Chen, Y. Liu, H. Sun, Physics-informed learning of governing equations from scarce data, *Nature communications* 12 (1) (2021) 6136.
- [14] M. Raissi, Deep hidden physics models: Deep learning of nonlinear partial differential equations, *The Journal of Machine Learning Research* 19 (1) (2018) 932–955.
- [15] M. Raissi, P. Perdikaris, G. E. Karniadakis, Physics-informed neural networks: A deep learning framework for solving forward and inverse problems involving nonlinear partial differential equations, *Journal of Computational physics* 378 (2019) 686–707.
- [16] G. E. Karniadakis, I. G. Kevrekidis, L. Lu, P. Perdikaris, S. Wang, L. Yang, Physics-informed machine learning, *Nature Reviews Physics* 3 (6) (2021) 422–440.
- [17] M.-F. Ng, J. Zhao, Q. Yan, G. J. Conduit, Z. W. Seh, Predicting the state of charge and health of batteries using data-driven machine learning, *Nature Machine Intelligence* 2 (3) (2020) 161–170.
- [18] B. Jiang, W. E. Gent, F. Mohr, S. Das, M. D. Berliner, M. Forsuelo, H. Zhao, P. M. Attia, A. Grover, P. K. Herring, et al., Bayesian learning for rapid prediction of lithium-ion battery-cycling protocols, *Joule* 5 (12) (2021) 3187–3203.
- [19] C. Lin, J. Xu, J. Hou, D. Jiang, Y. Liang, X. Zhang, E. Li, X. Mei, A fast data-driven battery capacity estimation method under non-constant current charging and variable temperature, *Energy Storage Materials* 63 (2023) 102967.
- [20] J. Zhu, Y. Wang, Y. Huang, R. Bhushan Gopaluni, Y. Cao, M. Heere, M. J. Mühlbauer, L. Mereacre, H. Dai, X. Liu, et al., Data-driven capacity estimation of commercial lithium-ion batteries from voltage relaxation, *Nature communications* 13 (1) (2022) 2261.

- [21] F. Xia, K. Wang, J. Chen, State of health and remaining useful life prediction of lithium-ion batteries based on a disturbance-free incremental capacity and differential voltage analysis method, *Journal of Energy Storage* 64 (2023) 107161.
- [22] M. Lin, C. Yan, W. Wang, G. Dong, J. Meng, J. Wu, A data-driven approach for estimating state-of-health of lithium-ion batteries considering internal resistance, *Energy* 277 (2023) 127675.
- [23] A. Geslin, B. van Vlijmen, X. Cui, A. Bhargava, P. A. Asinger, R. D. Braatz, W. C. Chueh, Selecting the appropriate features in battery lifetime predictions, *Joule* 7 (9) (2023) 1956–1965.

REVIEWERS' COMMENTS

Reviewer #1 (Remarks to the Author):

The authors have satisfactorily answered all my questions. I have no additional comments to make.

Reviewer #2 (Remarks to the Author):

The authors have addressed my questions. I don't have any additional comments

Reviewer #3 (Remarks to the Author):

All comments were addressed in detail. The explanation of the procedure is now broader, which makes the paper easier to understand. Together with the changes with reference to Reviewer 1, readability has certainly improved significantly. We therefore support the publication of the manuscript.

Reviewer #3 (Remarks on code availability):

Good readability of the code, comments translated to english.

Reviewer #4 (Remarks to the Author):

Reviewer #4 (Remarks on code availability):

Clear and understandable